# Precise temporal control of neuroblast migration through combined regulation and feedback of a Wnt receptor

Erik S Schild[1], Shivam Gupta[2], Clément Dubois[3], Euclides E Fernandes Póvoa[1], Marie-Anne Félix[3], Andrew Mugler[2,4]*, Hendrik C Korswagen[1,5]*

[1]Hubrecht Institute, Royal Netherlands Academy of Arts and Sciences and University Medical Center Utrecht, Utrecht, Netherlands; [2]Department of Physics and Astronomy, Purdue University, West Lafayette, United States; [3]Institut de Biologie de l'Ecole Normale Supérieure, CNRS, INSERM, Paris, France; [4]Department of Physics and Astronomy, University of Pittsburgh, Pittsburgh, United States; [5]Institute of Biodynamics and Biocomplexity, Department of Biology, Utrecht University, Utrecht, Netherlands

*For correspondence:
andrew.mugler@pitt.edu (AM);
r.korswagen@hubrecht.eu (HCK)

Competing interest: The authors declare that no competing interests exist.

## Abstract
Many developmental processes depend on precise temporal control of gene expression. We have previously established a theoretical framework for regulatory strategies that can govern such high temporal precision, but experimental validation of these predictions was still lacking. Here, we use the time-dependent expression of a Wnt receptor that controls neuroblast migration in *Caenorhabditis elegans* as a tractable system to study a robust, cell-intrinsic timing mechanism in vivo. Single-molecule mRNA quantification showed that the expression of the receptor increases non-linearly, a dynamic that is predicted to enhance timing precision over an unregulated, linear increase in timekeeper abundance. We show that this upregulation depends on transcriptional activation, providing in vivo evidence for a model in which the timing of receptor expression is regulated through an accumulating activator that triggers expression when a specific threshold is reached. This timing mechanism acts across a cell division that occurs in the neuroblast lineage and is influenced by the asymmetry of the division. Finally, we show that positive feedback of receptor expression through the canonical Wnt pathway enhances temporal precision. We conclude that robust cell-intrinsic timing can be achieved by combining regulation and feedback of the timekeeper gene.

## Editor's evaluation
This paper deals with an important unsolved problem in developmental biology: how cells execute their dynamics at the right time. The study combines compelling quantitative single-cell and single-transcript experiments with genetic perturbations and computational modelling and provides important insights into how the timing of transcription is regulated.

## Introduction
Timing plays a central role in development, coordinating processes ranging from cell division (*Tyson and Novak, 2008*) and differentiation (*Ray et al., 2022*), to the complex segmentation (*Dequéant and Pourquié, 2008*) and limb development of vertebrate embryos (*Pickering et al., 2018*). To measure time, biological clocks generally utilize a component that increases or decreases in activity or abundance (a timer) or cycles through a high and low state (an oscillator) and triggers a response when

a specific threshold is crossed (*Gliech and Holland, 2020*). An example of such a biological clock is the time-dependent expression of Neuropilin-1 in retinal ganglion cells (RGCs), which influences the trajectory of the outgrowing RGC axons (*Baudet et al., 2011*; *Campbell et al., 2001*). Here, the clock mechanism consists of a transcriptional repressor (CoREST) that gradually decreases in abundance as a result of miRNA dependent inhibition and releases Neuropilin-1 expression once it reaches a specific threshold. With its dependence on protein production, degradation and molecular interactions between proteins, such a timekeeping mechanism is inherently noisy, raising questions on the regulatory strategies that biological clocks have evolved to increase precision.

In clocks that act across multiple cells, robustness can be increased through averaging, synchronization through intercellular communication (*Dequéant and Pourquié, 2008*; *Oates, 2020*) or entrainment using external signals (*Aydogan et al., 2020*; *Bell-Pedersen et al., 2005*). The cell intrinsic mechanisms that control the timekeeping mechanism itself are, however, still poorly understood. Mathematical modeling has provided insight into regulatory circuits that can enhance timing precision. One mechanism for temporal regulation is the constant production of a stable timer molecule, which leads to a gradual accumulation and activation of the time-dependent response once a specific abundance threshold is reached. Modeling has shown that such a linear increase is more precise than a strategy where the timer molecule positively regulates its own expression, as such autoregulation also amplifies the noise (*Ghusinga et al., 2017*). Using a first-passage time approach, we have recently shown that a regulated strategy where the timekeeping molecule increases non-linearly over time can increase precision (*Gupta et al., 2018*). Such regulation can be mediated through an accumulating transcriptional activator or a gradually decreasing transcriptional repressor that controls the expression of the timekeeper molecule in a threshold-dependent manner, and we showed this to be robust to additional effects such as cell division and bursts in transcription (*Gupta et al., 2018*). Moreover, in contrast to autoregulation on its own, we found that precision could be further increased when the regulator was combined with autoregulation of the timekeeper (*Gupta et al., 2020*). It is not known, however, if this predicted regulatory strategy is utilized by biological clocks in vivo.

The migration of the *Caenorhabditis elegans* QR neuroblast descendants is controlled in a time-dependent fashion (*Mentink et al., 2014*) and provides a tractable system to study a biological clock at the single-cell level in vivo. QR is born in the mid-body as a sister cell of the hypodermal seam cell V5 and divides at the beginning of the first larval stage into an anterior daughter cell called QR.a and a posterior daughter QR.p (*Figure 1A*). Both daughter cells migrate a relatively long distance toward the anterior and divide at highly stereotypic positions to generate descendants that differentiate into neurons (*Sulston and Horvitz, 1977*). While QR.a divides only once to generate QR.ap and an apoptotic sister cell QR.aa, QR.p first divides into QR.pa and an apoptotic sister cell QR.pp. QR.pa continues with a short-range migration toward the anterior but then stops to divide into QR.paa and QR.pap, which migrate dorsally and ventrally to occupy their final positions, where they differentiate into a mechanosensory neuron and an interneuron, respectively. Compared to other neurons that undergo long-range migration in *C. elegans*, the final position of the QR.p descendants was found to be robust to stochastic variation (*Dubois et al., 2021*).

The anterior migration of QR.p and QR.pa is dependent on Wnt signaling (*Mentink et al., 2014*; *Rella et al., 2021*). Wnt proteins are an evolutionarily conserved family of extracellular ligands that can signal through a canonical, β-catenin-dependent pathway to induce the expression of specific target genes, or through β-catenin-independent, non-canonical pathways that directly influence cytoskeletal dynamics (*Angers and Moon, 2009*; *Clevers and Nusse, 2012*). We have previously shown that the anterior migration of QR.p and QR.pa is mediated through parallel acting non-canonical Wnt pathways that separately control speed and polarity (*Mentink et al., 2014*). Once QR.pa reaches its final position, migration is stopped through activation of canonical, BAR-1/β-catenin-dependent Wnt signaling, which induces the expression of target genes that inhibit migration (*Rella et al., 2021*). This switch in signaling response is mediated through the Wnt receptor MIG-1/Frizzled (*Mentink et al., 2014*). Quantitative in vivo measurement of *mig-1* expression using single-molecule FISH (smFISH) showed that *mig-1* expression is rapidly upregulated in QR.pa, and transgenic rescue experiments demonstrated that *mig-1* is both necessary and sufficient for migration termination (*Mentink et al., 2014*). Importantly, we found that this upregulation is not dependent on positional information but that the expression of *mig-1* is temporally regulated. Thus, in mutants in which the QR descendants migrate at a slower speed, *mig-1* is still expressed at the same time but at a more posterior position,

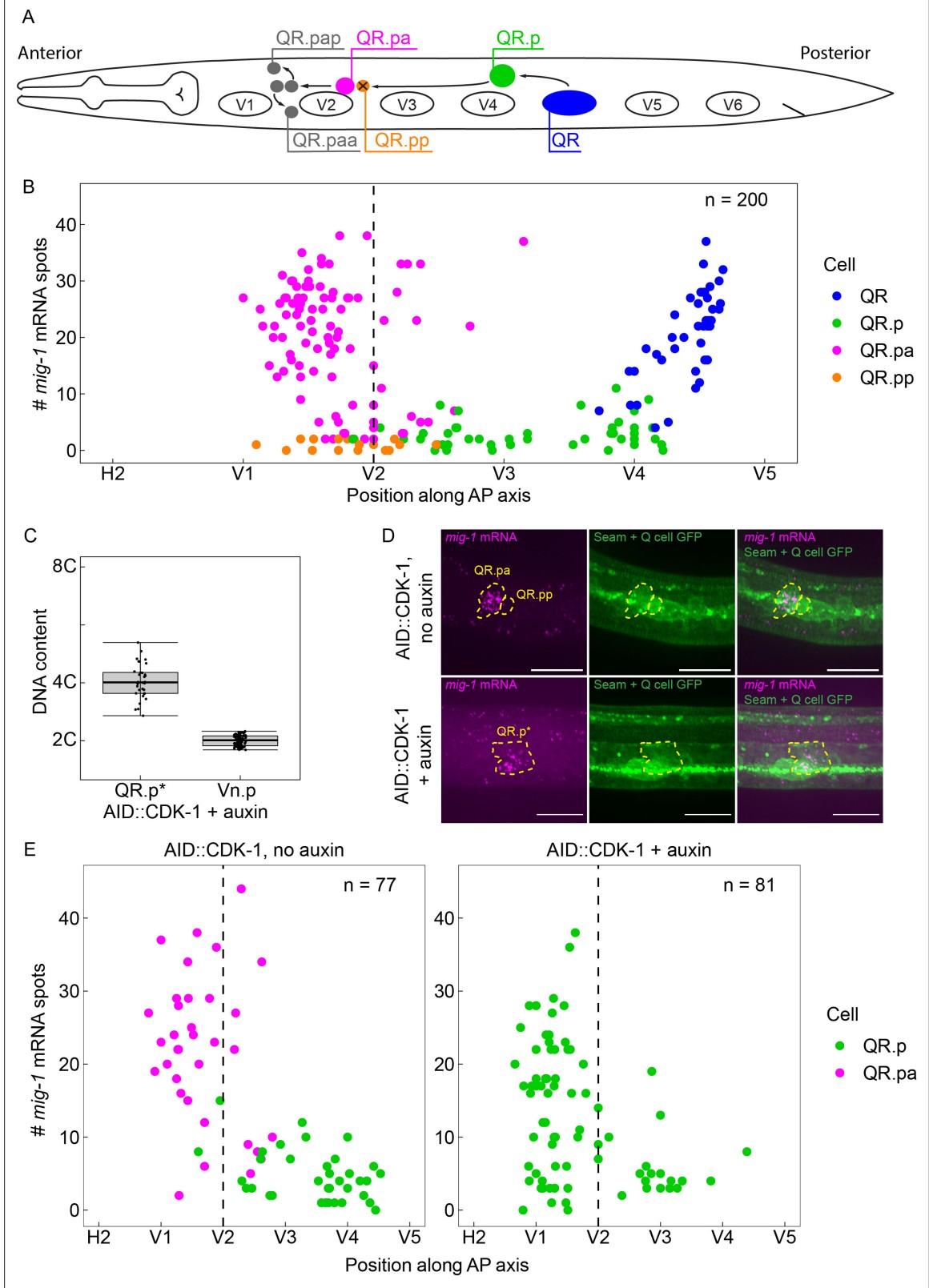

**Figure 1.** The time-dependent expression of *mig-1* is independent of QR.p division. (**A**) Schematic representation of the QR lineage and the migration of the QR descendants. V1–V6 are seam cells, used as landmarks to determine the position of QR and its descendants. For clarity, the QR.a branch of the QR lineage is not represented. (**B**) Single-molecule FISH (smFISH) quantification of *mig-1* expression in QR and its descendants relative to the position of the seam cells H2 to V5. n=200 from 11 independent experiments, with three replicates each. (**C**) DNA content in division-blocked QR.p

*Figure 1 continued on next page*

*Figure 1 continued*

(QR.p*), relative to 2C (Vn.p) seam cells. The DNA content of QR.p* does not significantly deviate from 4C (p=0.31, n=32, from five independent experiments, with three replicates each), indicating the cells do not undergo an additional S-phase after the division is blocked. Whiskers represent 1.5x interquartile range. (**D**) Representative images of *mig-1* smFISH spots in *cdk-1*(*hu277*[*AID::cdk-1*]). Top: control without auxin and bottom: QR.p division is blocked in the presence of auxin. Scale bar is 10 μm. Green Fluorescent Protein (GFP) is expressed in the seam cells and the Q neuroblast descendants (*heIs63*; *Wildwater et al., 2011*). (**E**) smFISH quantification of *mig-1* expression in QR and its descendants relative to the position of the seam cells H2 to V5 in *cdk-1*(*hu277*[*AID::cdk-1*]) in the absence (left) and presence (right) of auxin. Data are from seven independent experiments, with three replicates each.

The online version of this article includes the following figure supplement(s) for figure 1:

**Figure supplement 1.** *mig-1* single-molecule FISH (smFISH) spot counts from *Figure 1B* in the different replicate experiments.

while in animals with faster migrating QR descendants, *mig-1* is expressed at a more anterior position. These results demonstrate that *mig-1* expression is regulated through a timing mechanism and that the final position of the QR descendants is determined by the speed of migration and the time at which *mig-1* expression is induced. Consistent with such a timing mechanism, the final QR descendants localize to a more anterior position in mutants with a shorter body size and at a more posterior position in animals with a longer body size, although it should be noted that some compensation takes place at the level of migration speed (*Dubois et al., 2021*).

Here, we combine mutant analysis and mathematical modeling to examine the timekeeping mechanism that controls *mig-1* expression in the QR lineage. We show that the rapid upregulation of *mig-1* expression in QR.pa is dependent on transcriptional activation, supporting a model in which the non-linear increase in *mig-1* expression is mediated through threshold-crossing of an accumulating transcriptional activator. We find that this timing mechanism acts across the division of QR.p but is influenced by the asymmetry of the division. Finally, we show that positive feedback of *mig-1* expression through the canonical Wnt pathway decreases temporal variability. We conclude that robust timing of gene expression can be achieved by combining an accumulating transcriptional activator with feedback regulation of the timekeeper molecule.

## Results

### Temporal regulation of *mig-1* expression in the QR lineage is independent of cell division and cell cycle progression

During the anterior migration of the QR descendants, *mig-1* expression is low in QR.p but is strongly upregulated in its daughter QR.pa (*Mentink et al., 2014*; *Figure 1B*), where it triggers the canonical Wnt signaling response that is necessary to stop migration (*Rella et al., 2021*). The presence of a cell division prior to the upregulation of *mig-1* expression – the timing of which is tightly controlled as part of the invariant cell lineage of *C. elegans* development (*Sulston and Horvitz, 1977*) – raises the question whether the two are mechanistically linked. To investigate whether the upregulation of *mig-1* expression is dependent on the division of QR.p (or subsequent cell cycle reentry in QR.pa), we blocked QR.p mitosis by conditionally depleting the M-phase regulator CDK-1 using the auxin inducible protein degradation system (*Zhang et al., 2015*). In this system, proteins tagged with an auxin inducible degron (AID) sequence are degraded in the presence of auxin and the F-box protein TIR1 (*Zhang et al., 2015*). We endogenously tagged *cdk-1* with the AID sequence (AID::CDK-1) using CRISPR/Cas9-mediated genome editing and specifically expressed TIR1 in the Q neuroblast lineage using the *egl-17* promoter (*Branda and Stern, 2000*). Since continuous exposure to auxin would block all divisions in the QR lineage, auxin was only applied from the stage at which QR has completed its division (300–345 min after hatching). We found that this efficiently inhibited the subsequent division of QR.p and its sister cell QR.a, as judged by the absence of their descendants (the apoptotic QR.pp and QR.aa cells and the neuronal QR.paa, QR.pap, and QR.ap cells).

Studies in yeast and mammalian cells have shown that loss of CDK1 not only inhibits mitosis but also cell cycle reentry and DNA replication (*Coudreuse and Nurse, 2010*). To investigate if the cell cycle is similarly blocked in the undivided QR.p cells, we examined DNA content by measuring total 4′,6-diamidino-2-phenylindole (DAPI) fluorescence at a time point at which the daughter of QR.p (QR.pa) would normally have divided into the final descendants QR.paa and QR.pap (8–9 hr after hatching) and normalizing to total DAPI fluorescence of the Vn.p (seam) cell nuclei, which have two unreplicated

sets of chromosomes (2C) at this stage in L1 larval development (*Hedgecock and White, 1985*) and have a similarly sized nucleus. When mitosis and subsequent cell cycle progression is blocked by CDK-1 depletion, the DNA content of the replicated but non-divided QR.p nucleus is expected to be 4C. However, if cycling and DNA replication continues, a DNA content of 8C or higher is expected. As shown in *Figure 1C*, we observed no significant deviation from a DNA content of 4C. These results show that CDK-1 depletion inhibits mitosis as well as cell cycle reentry in the undivided QR.p cells.

Next, we measured *mig-1* expression in the AID::CDK-1 strain using smFISH (*Ji et al., 2013*). In the absence of auxin, *mig-1* expression was not significantly different from animals containing wild type CDK-1 (*Figure 1B, D and E*). Importantly, however, we found that in the presence of auxin, upregulation of *mig-1* expression in undivided QR.p cells occurs at a similar time and range of expression as in the control animals (*Figure 1D and E*). We conclude that the temporal regulation of *mig-1* expression is independent of the cell cycle and QR.p division.

## The early and late phases of *mig-1* expression are independently regulated

The expression of *mig-1* in the QR lineage occurs in two phases (*Mentink et al., 2014*; *Figure 1B*): an early phase in which it is expressed in the QR neuroblast, followed by a rapid decrease during QR polarization (*Ji et al., 2013*) and low expression in QR.p until *mig-1* is upregulated again in QR.pa (the late phase). To gain insight into the transcriptional regulation of *mig-1*, we made the assumption that cis-regulatory elements that control the expression of *mig-1* are under stabilizing selection and therefore conserved across species (*Gordân et al., 2010*; *Harbison et al., 2004*; *Nitta et al., 2015*). By comparing the upstream region and first intron of *mig-1* in 23 *Caenorhabditis* species, we found eight conserved 25 bp motifs that are located in four closely linked pairs (*Figure 2—figure supplement 1*, *Figure 3—figure supplement 1*).

Two pairs of motifs are located in the first intron (motif pair A and B; *Figure 2A*, *Figure 2—figure supplement 1*). To test if these motifs are required for the expression of *mig-1*, we independently deleted each of the two pairs using CRISPR/Cas9-mediated genome editing and quantified the number of *mig-1* transcripts in QR and its descendants using smFISH. While deletion of motif pair B had no effect on *mig-1* expression, deletion of motif pair A strongly reduced the early expression of *mig-1* in the QR neuroblast (*Figure 2B and C*). The late expression of *mig-1* in QR.pa was, however, not affected by deletion of this motif pair. Moreover, using the final position of the QR.pa daughter cell QR.pap as a measure of total migration distance, we found that the final position of the QR descendants was not significantly different from control animals (*Figure 2—figure supplement 2*). This is consistent with the observation that only the late phase of *mig-1* expression is required for correct QR descendant migration (*Mentink et al., 2014*). Taken together, these results show that intron motif pair A is necessary for the early but not the late expression of *mig-1* and support the notion that the two phases are independently regulated.

## A dynamical model of the temporal regulation of *mig-1* expression

Taking into account that the late phase of *mig-1* expression is independent of cell division and the early phase of expression, we developed a dynamical model of a cell-intrinsic timer mechanism driving *mig-1* transcription. Since the speed of QR.p migration – which crosses most of the distance covered by the QR descendants – is roughly constant (*Mentink et al., 2014*), we treated distance along the anteroposterior axis as proportional to time. To account for the decrease (QR), followed by the increase (QR.pa), in the *mig-1* amount over time, we hypothesized that *mig-1* mRNA is continuously degraded and upregulated by a component with its own dynamics: either a transcriptional activator that increases in time or a repressor that decreases in time (*Gupta et al., 2018*). Thus, the dynamics of *mig-1* mRNA number $m$ are described by $dm/dt = -\nu m + F(t)$, where $\nu$ is the degradation rate, and $F(t)$ accounts for the regulation. For the activator model, we set $F(t) = \alpha a^H / \left( a^H + K^H \right)$, where the activator molecule number increases with rate $k$ according to $a(t) = kt$. For the repressor model, we set $F(t) = \alpha K^H / \left( r^H + K^H \right)$, where the repressor molecule number decreases with rate μ from initial value $r_0$ according to $r(t) = r_0 e^{-\mu t}$. In both models, $\alpha$ is the maximal production rate of *mig-1*, $H$ is the Hill coefficient, and $K$ is the half-maximal value of the regulator. Fitting either model to the *mig-1* expression data (see Materials and methods) gave good agreement (*Figure 2D*, *Figure 2—figure supplement 3*). Consistent with the intron motif pair A deletion data, we found that lowering the

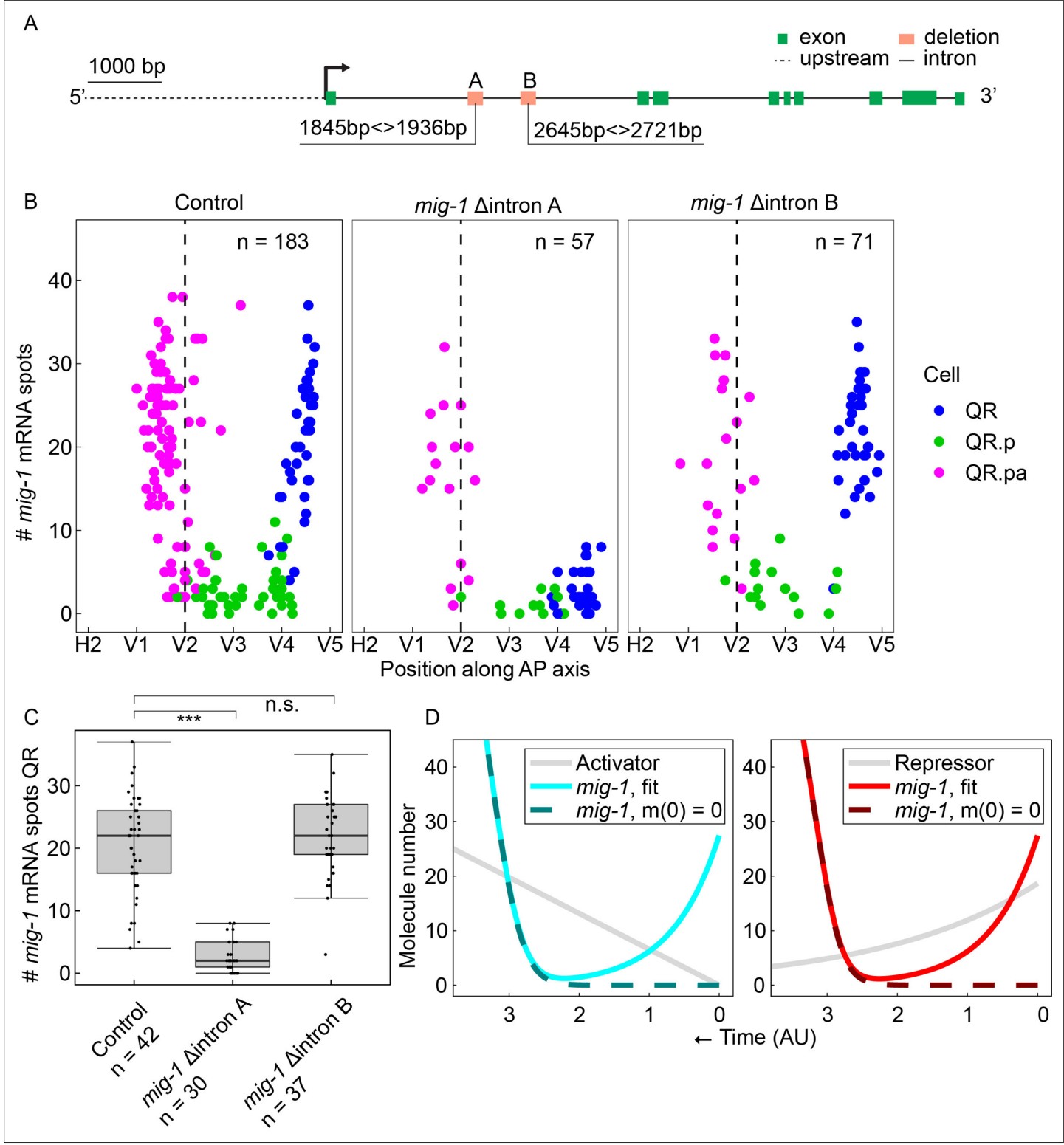

**Figure 2.** A conserved region in the first intron of *mig-1* controls the early but not the late phase of *mig-1* expression. (**A**) Schematic representation of the two conserved intron regions that were deleted from the *mig-1* locus. Green rectangles are exons, the gray dashed line is the upstream region, and black solid lines are introns. The orange rectangles represent the deleted regions that contain intron motif pair A (Δintron A) and intron motif pair B (Δintron B). Data are from three independent experiments, with three replicates each. (**B**) Single-molecule FISH (smFISH) quantification of *mig-1* expression in QR and its descendants relative to the position of the seam cells H2 to V5 in control and the intron deletion mutants *hu295* (intron deletion A) and *hu299* (intron deletion B). AP, anteroposterior. (**C**) Quantification of *mig-1* smFISH spots in QR (***p<0.0001, n=30, Welch's t-test). Whiskers

*Figure 2 continued on next page*

*Figure 2 continued*

represent 1.5x interquartile range. (**D**) Mathematical model for the *mig-1* dynamics $m\left(t\right)$ regulated by an increasing activator (left) or decreasing repressor (right), fit to the control data (see Materials and methods and Figure 2-figure supplement 3). The sequences of the intron motifs in 23 different *Caenorhabditis* species are in ***Figure 2—figure supplement 1***. ***Figure 2—figure supplement 2*** shows that deletion of the intron motif pairs does not significantly change the final position of QR.pap.

The online version of this article includes the following figure supplement(s) for figure 2:

**Figure supplement 1.** Conserved motifs in the first intron of *mig-1* in 23 different *Caenorhabditis* species.

**Figure supplement 2.** Deletion of intron motif pair A and B does not significantly change the final position of QR.pap.

**Figure supplement 3.** Overlay of fits from ***Figure 2D*** on *mig-1* single-molecule FISH (smFISH) data from ***Figure 2B***.

initial *mig-1* amount ($m\left(0\right) = 0$) does not affect the later upregulation dynamics (***Figure 2D***). Therefore, the model captures the observation that early and late *mig-1* expression dynamics are independently regulated.

## The late phase of *mig-1* expression is dependent on two conserved, upstream activating cis-regulatory elements

The increasing transcriptional activator model and the decreasing transcriptional repressor model make contrasting predictions if the regulator is removed. Removing the activator does not influence *mig-1* expression at early timepoints and only prevents upregulation of *mig-1* at later timepoints (***Figure 3A***). Conversely, removing the repressor causes premature upregulation at early timepoints (***Figure 3A***). To gain insight into the transcriptional regulation of the late phase of *mig-1* expression and to test the predictions of the activator and repressor models, we focused on conserved motifs in the *mig-1* upstream region.

Two pairs of conserved, closely spaced motifs were identified in a 3 kb region upstream of the *mig-1* locus (***Figure 3B***, ***Figure 3—figure supplement 1***): a distal pair (#1) at 2781–2571 bp and a proximal pair (#2) at 267–137 bp from the start of the *mig-1* coding sequence. Using CRISPR/Cas9-mediated genome editing, we individually deleted each of the motif pairs and also created a double mutant in which both pairs are deleted simultaneously. None of the deletions had an effect on the early phase of *mig-1* expression in the QR neuroblast (***Figure 3C***). However, we found that in the double deletion mutant, the late phase of *mig-1* expression in QR.pa was completely lost (***Figure 3C***). A similar but less penetrant reduction was observed in the single motif pair 2 deletion, while in the motif pair 1 deletion a minor fraction of QR.pa cells failed to upregulate *mig-1* expression (***Figure 3C***), a difference that did not result in a statistically significant difference in the mean expression of *mig-1*. Consistent with the role of the late phase of *mig-1* expression in terminating the anterior migration of QR.pa, the average position of QR.pa (p<0.0001 for the single and double deletions) and the total migration of the QR descendants (as determined from the final position of QR.pap) was shifted anteriorly (***Figure 3—figure supplement 2***). As expected, this effect on total migration distance was most pronounced in the double deletion and the single motif pair 2 deletion, but there was also significant overmigration in the motif pair 1 deletion mutant, indicating that the mild reduction in *mig-1* expression observed in this mutant is still sufficient to affect the final position of the QR descendants. Taken together, these results demonstrate that the two motif pairs represent partially redundant cis-regulatory elements that are required for the late phase of *mig-1* expression and provide further support for the conclusion that the early and late phases of the expression are independently regulated.

Importantly, the loss of the late phase of *mig-1* expression – instead of premature expression earlier in the lineage – is in agreement with the activator model, not the repressor model, of *mig-1* regulation (***Figure 3A***).

## Asymmetry of the QR.p division is required for upregulation of *mig-1* expression in QR.pa

The division of QR.p that precedes the upregulation of *mig-1* expression in QR.pa is asymmetric in the size and fate of its daughter cells: the large anterior daughter cell QR.pa remains a neuroblast, while the small posterior daughter cell QR.pp undergoes apoptosis and is rapidly engulfed and degraded

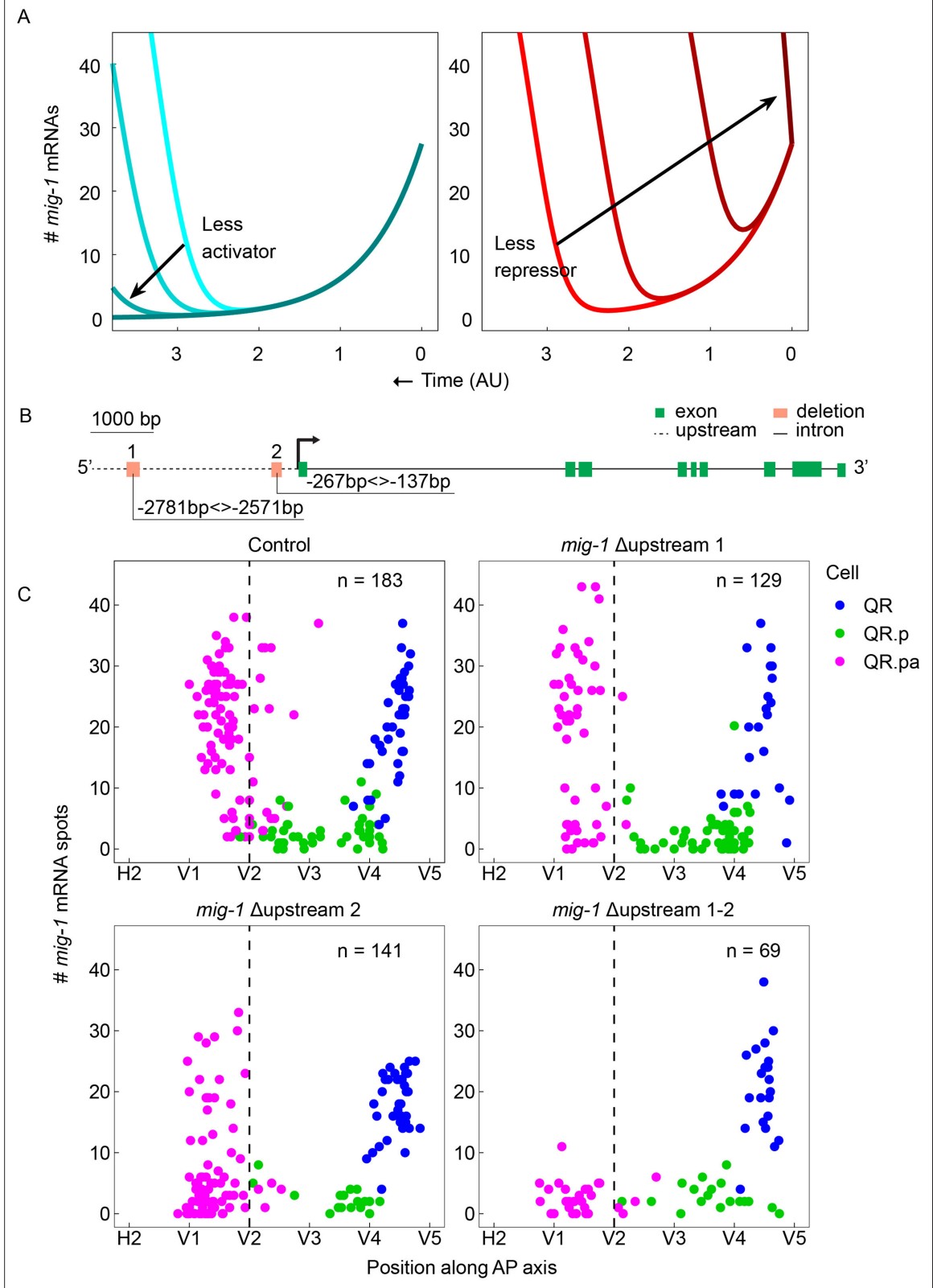

**Figure 3.** Two conserved regions upstream of *mig-1* control the late but not the early phase of *mig-1* expression. (**A**) Mathematical model with decreasing activator production rates $k = [6.6, 5.6, 4.6, 0]$ (left) or decreasing initial repressor numbers $r_0 = [19, 13, 7, 0]$ (right). (**B**) Schematic representation of the two conserved regions in the *mig-1* upstream sequence. Green rectangles are exons, the gray dashed line is the upstream region, and black solid lines are introns. The orange rectangles represent the deleted regions that contain upstream motif pair #1 (Δupstream 1) and

*Figure 3 continued on next page*

*Figure 3 continued*

upstream motif pair #2 (Δupstream 2). (**C**) Single-molecule FISH (smFISH) quantification of *mig-1* expression in QR and its descendants relative to the position of the seam cells H2 to V5 in control and deletion mutants of the two regions upstream of *mig-1*. None of the deletions affect the early phase of *mig-1* expression in the QR neuroblast (*hu314* Δupstream 1), p=0.59, n=22; *hu315* (Δupstream 2), p=0.07, n=40; *hu335* (Δupstream 1–2), p=0.89, n=22, Welch's t-test. The upregulation of *mig-1* expression in QR.pa is not significantly changed in Δupstream 1 (p=0.44, n=51) but is significantly reduced in Δupstream 2 (p<0.0001, n=83) and abolished in the double Δupstream 1–2 deletion (p<0.0001, n=29). Data are from eight independent experiments with three replicates for Δupstream 1 and 2 and three independent experiments with three replicates for the double Δupstream 1–2 deletion. The sequences of the upstream motifs in 23 different *Caenorhabditis* species are in *Figure 3—figure supplement 1*. *Figure 3—figure supplement 2* shows the effect of the upstream motif pair deletions on the final position of QR.pap.

The online version of this article includes the following figure supplement(s) for figure 3:

**Figure supplement 1.** Conserved motifs in the upstream region of *mig-1* in 23 different *Caenorhabditis* species.

**Figure supplement 2.** Deletion of upstream motif pair 1 and 2 induces significant overmigration of QR.pap.

by neighboring cells (*Sulston and Horvitz, 1977*). Although we found that the division per se is not necessary for the late phase of *mig-1* expression, the size difference of the daughter cells and potentially unequal inheritance of the *mig-1* activator could contribute to the temporal regulation of *mig-1* expression.

To investigate the role of the asymmetric division of QR.p in *mig-1* regulation, we first compared *mig-1* expression in the two daughter cells using a null mutation in the essential cell death regulator *ced-3* to prevent apoptosis of QR.pp (*Ellis and Horvitz, 1986*). As expected, loss of *ced-3* did not affect the upregulation of *mig-1* expression in QR.pa (*Figure 4A*). In contrast, the number of *mig-1* transcripts that could be detected in the surviving QR.pp cells remained low. These results indicate that *mig-1* is differently regulated in the two daughter cells. Next, we investigated if disrupting the asymmetry of the QR.p division affects the expression of *mig-1* in QR.pa and QR.pp. A key regulator of asymmetric neuroblast division is the serine/threonine kinase PIG-1/MELK (*Cordes et al., 2006*), which controls the size difference of the QR.p daughters by inducing posterior displacement of the mitotic spindle (*Chien et al., 2013*) and may also contribute to the asymmetric segregation of cell fate determinants (*Chien et al., 2013*). As expected from these previous studies, loss of *pig-1* resulted in a variable defect in cell size asymmetry, with a significant reduction in the mean cell size difference between QR.pa and QR.pp (*Figure 4B*). smFISH analysis showed that *mig-1* expression in QR.pa was significantly decreased under these conditions (*Figure 4A*). However, there was no corresponding increase in the number of *mig-1* transcripts in QR.pp, and no evidence of active transcription – which is visible as one or two bright smFISH spots in the nucleus (*Ji et al., 2013*) – was observed. To examine if *mig-1* expression in QR.pa correlates with the level of cell size asymmetry, we compared the size ratio of QR.pa/QR.pp pairs with *mig-1* expression in QR.pa. As shown in *Figure 4C*, we found a moderate correlation, with QR.pa cells from less asymmetric pairs showing lower *mig-1* expression than QR.pa cells from more asymmetric pairs.

To understand the effect of division asymmetry on *mig-1* expression, we returned to our activator-based mathematical model. First, we incorporated QR.p division into the model at a time inferred from the data (see Materials and methods). Because no active transcription of *mig-1* was observed in QR.pp experimentally, we set the *mig-1* production rate to zero in QR.pp. Despite changes in the overall cell size, we experimentally observed that the nuclear size of QR.pa is not significantly different from that of its parent QR.p (*Figure 4D*). Since the activator is likely a transcription factor and therefore acts in the nucleus, the nuclear concentration of the activator in QR.pa should thus depend on its molecule number but not the cell size. Therefore, as before, we assumed that the production of *mig-1* in QR.pa is dependent on the number of molecules, and not the concentration, of the activator. Finally, because the nuclear envelope breaks down before division, we assumed that at division, both the activator and preexisting *mig-1* transcripts are distributed according to the relative sizes of QR.pa and QR.pp. Thus, at the division time, both the activator and *mig-1* transcript numbers in QR.p are multiplied by a factor of $f$ in QR.pa and $(1 - f)$ in QR.pp, where $f = \rho/(1 + \rho)$ for a given QR.pa/QR.pp size ratio $\rho$. The solution to this model with parameters fit to the control data (see Materials and methods and *Figure 4—figure supplement 1*) and the wild type mean $\rho = 3.5$ is shown in *Figure 4E* (dark colors). We observed that decreasing the QR.pa/QR.pp size ratio leads to reduced *mig-1* levels in the QR.pa dynamics (*Figure 4E*, light colors), consistent with the experiments (*Figure 4A*). Correspondingly, plotting the *mig-1* level as a function of the size ratio gave a positive dependency in the

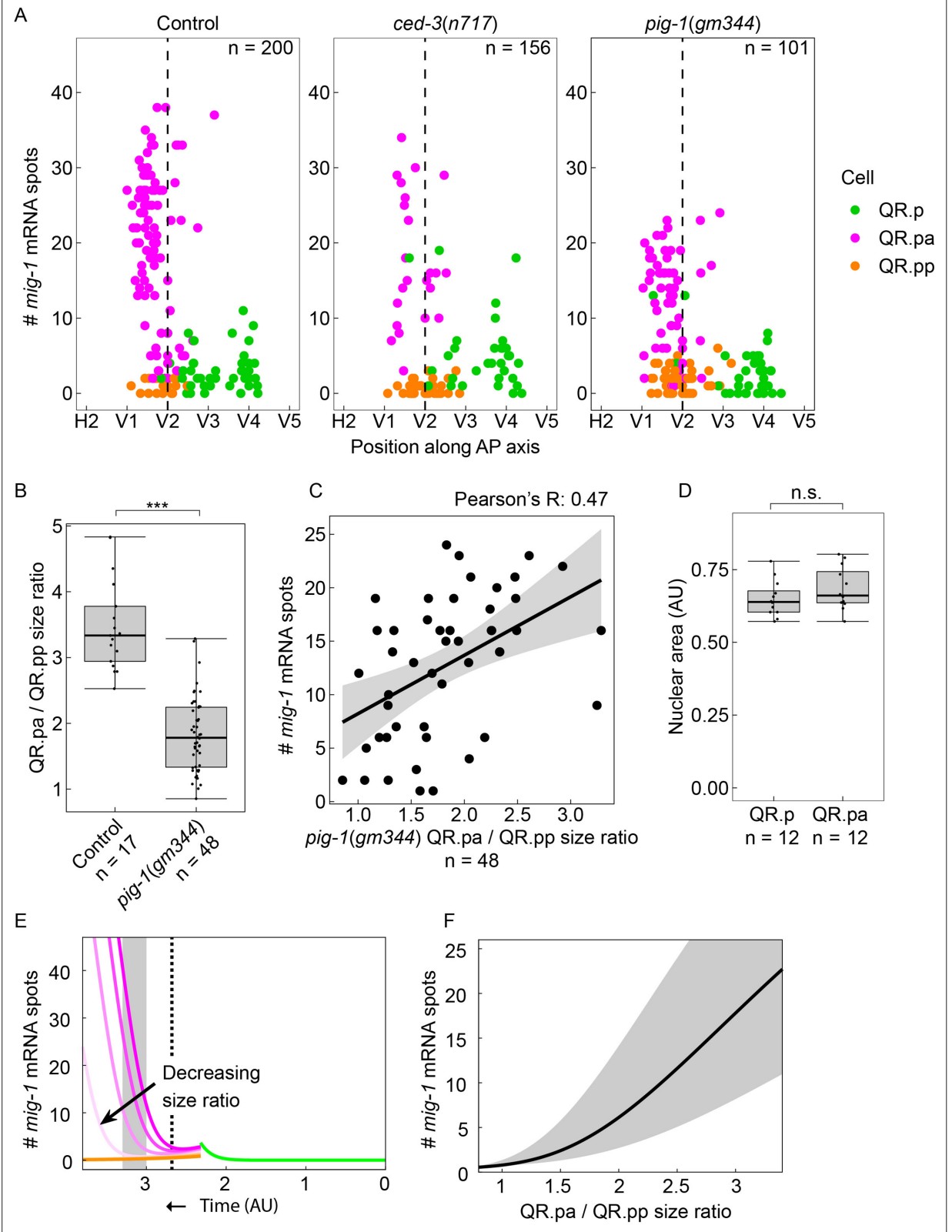

**Figure 4.** QR.p division asymmetry influences *mig-1* expression dynamics in QR.pa. (**A**) Single-molecule FISH (smFISH) quantification of *mig-1* expression in QR and its descendants relative to the position of the seam cells H2 to V5 in control, *ced-3(n717)*, and *pig-1(gm344)* mutants. *mig-1* expression in QR.pa is not significantly different between control and *ced-3* mutants (p=0.36, n=34, Welch's t-test) but is significantly reduced in *pig-1* mutants (p<0.0001, n=56). Data are from four independent experiments with three replicates each. (**B**) The size ratio of QR.pa and QR.pp in control

*Figure 4 continued on next page*

*Figure 4 continued*

(*heIs63*) and *pig-1(gm344)* mutants (***p<0.0001, n=48, from three independent experiments with three replicates each). Whiskers represent 1.5x interquartile range. (**C**) The QR.pa/QR.pp size ratio and *mig-1* expression show a moderately positive correlation in *pig-1(gm344)* mutants (Pearson correlation, R=0.47, n=48). (**D**) Nuclear size is not significantly different in QR.p and its daughter QR.pa in wild type control animals (p=0.22, n=12, Welch's t-test). Whiskers represent 1.5x interquartile range. (**E**) Mathematical model incorporating cell division, fit to control data (see Materials and methods and *Figure 4—figure supplement 1*) with wild type QR.pa/QR.pp size ratio $\rho = 3.5$ (dark colors) and for lowered size ratios $\rho = [2.7, 1.8, 1]$ (light colors). Dashed line: time corresponding to moment in QR lineage migration that seam cell V2 is reached. Gray box: time window corresponding to gray region in F. (**F**) Plot of *mig-1* molecule number in QR.pa vs. QR.pa/QR.pp size ratio in model at times between $t = 3$ and $t = 3.3$ (gray region) and at the midpoint time $t = 3.15$ (black line).

The online version of this article includes the following figure supplement(s) for figure 4:

**Figure supplement 1.** Overlay of fits from *Figure 4E* on *mig-1* single-molecule FISH (smFISH) data from *Figure 4A*.

model (*Figure 4F*), which is in agreement with the positive correlation in the experiments (*Figure 4C*). We conclude that the asymmetry of the QR.p division, with the concomitant difference in cell size and the share of the *mig-1* activator that is inherited by the two daughter cells, is required for upregulation of *mig-1* expression in QR.pa.

## Positive feedback regulation enhances timing precision of *mig-1* expression

Mathematical modeling of the regulator-based timekeeping mechanism has shown that temporal precision is enhanced when the regulated timekeeper molecule amplifies its own expression through positive feedback (*Gupta et al., 2020*), but the in vivo relevance of this predicted strategy is not known.

*mig-1* encodes a Frizzled Wnt receptor that triggers a BAR-1/β-catenin-dependent signaling pathway in QR.pa that induces the expression of target genes that inhibit migration (*Mentink et al., 2014*; *Rella et al., 2021*). Interestingly, mRNA sequencing showed that *mig-1* itself is among the genes that are significantly upregulated when BAR-1 signaling is prematurely activated in QR.p using a constitutively active, N-terminally truncated version of BAR-1 (ΔN-BAR-1; *Rella et al., 2021*). To examine this further, we quantified *mig-1* expression in *bar-1(ga80)* null mutants (*Eisenmann et al., 1998*) using smFISH. As shown in *Figure 5—figure supplement 1*, *mig-1* expression in QR.pa was significantly reduced in the absence of BAR-1 signaling. These results support the notion that MIG-1 stimulates its own expression through BAR-1/β-catenin-dependent positive feedback.

Next, we investigated whether positive feedback regulation of *mig-1* expression influences timing precision. Because BAR-1 signaling inhibits QR.pa migration (*Mentink et al., 2014*), the mean position of QR.pa is shifted anteriorly in the *bar-1* null mutant and posteriorly in ΔN-BAR-1 expressing animals (*Figure 5A*). We therefore used the coefficient of variation (CV, standard deviation divided by the mean) in position as a relative measure of timing noise across these different conditions. As shown in *Figure 5B*, the CV in the position at which QR.pa expresses *mig-1* at ≥25 transcripts was increased in both the *bar-1* null mutant and animals expressing ΔN-BAR-1.

To model the positive feedback, we considered three possibilities: the positive feedback (1) acts on *mig-1* independently of the activator, (2) acts on *mig-1* through the activator, or (3) both (*Figure 5—figure supplement 2A*). In all cases, we neglected cell division and *mig-1* degradation for simplicity, and we calculated the CV in the time at which $m$ reaches a threshold numerically from the master equation following our previous work (*Gupta et al., 2018*). In case 1, we found that removing the feedback has little effect on the CV because the removal does not affect the dynamics of the activator (*Figure 5—figure supplement 2*), and the timing precision of an activated species is only marginally improved by autoregulation (*Gupta et al., 2020*). In case 2, because removing the feedback affects the dynamics of the activator but not the activation function of *mig-1*, we found that the removal produces *mig-1* dynamics that are dramatically different from those observed in the *bar-1* null and ΔN-BAR-1 data (*Figure 5—figure supplement 2*). In case 3, because removing the feedback affects both the dynamics of the activator and the activation function of *mig-1*, we found that the removal produces linear *mig-1* dynamics whose CV is significantly larger than that with the feedback intact (*Figure 5B* inset and *Figure 5—figure supplement 2*). Since the *bar-1* null and ΔN-BAR-1 data are more linear (*Figure 5A*) and have a higher CV (*Figure 5B*) than the control data, we conclude that case

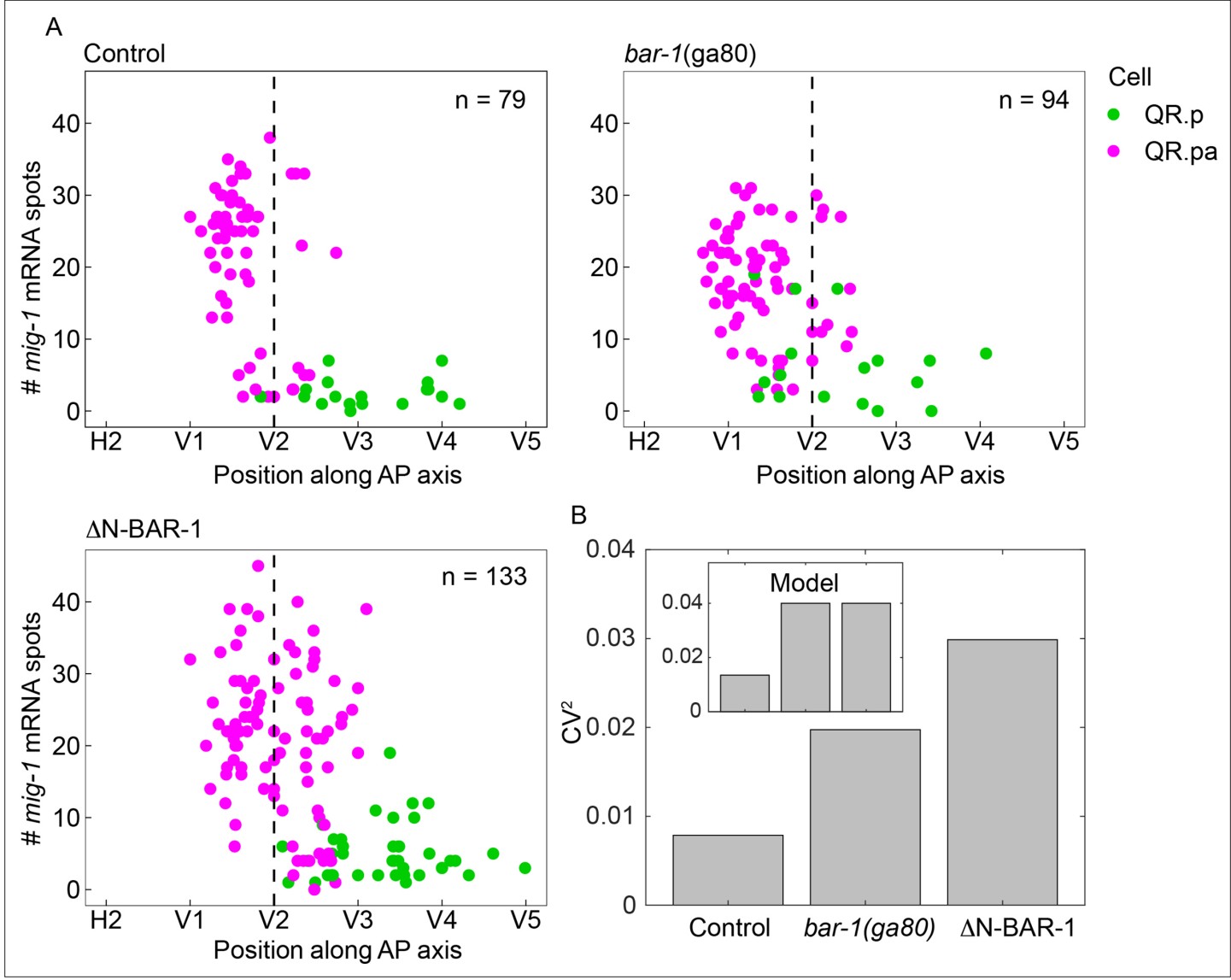

**Figure 5.** BAR-1/β-catenin dependent autoregulation of *mig-1* expression contributes to timing precision. (**A**) Single-molecule FISH (smFISH) quantification of *mig-1* expression in QR.p and QR.pa relative to the position of the seam cells H2 to V5 in control, *bar-1(ga80)* null mutants and animals expressing a constitutively active, N-terminally truncated version of BAR-1 (ΔN-BAR-1) in a *mab-5(gk670)* mutant background. Data are from four independent experiments with three replicates each. (**B**) Coefficient of variation (CV) squared of the position of QR.pa cells with ≥25 *mig-1* smFISH spots. Brown–Forsythe test for thresholds of 10–25 transcripts gives p=0.07–0.9 for control vs. *bar-1(ga80)*, and 3×10⁻⁶ to 0.008 for control vs. ΔN-BAR-1. Inset: CV² in the model in which positive feedback acts on *mig-1* both through the activator and independently of the activator (see ***Figure 5—figure supplement 2***, purple).

The online version of this article includes the following figure supplement(s) for figure 5:

**Figure supplement 1.** *mig-1* expression in QR.pa is reduced in *bar-1*/β-catenin null mutants.

**Figure supplement 2.** Modeling the effect of positive feedback on timing precision.

3 best describes the positive feedback. These results provide experimental support for the model prediction that positive feedback of a regulated timekeeper enhances timing precision.

## Discussion

Mathematical modeling of biological timekeeping mechanisms has provided insight into regulatory strategies that can achieve hightemporal precision (***Gupta et al., 2020***; ***Gupta et al., 2018***), but the relevance of these predicted mechanisms for in vivo timekeeping remains to be determined. Using

the time-dependent expression of the Wnt receptor gene *mig-1*/Frizzled in the *C. elegans* QR neuroblast lineage as a model system, we provide experimental evidence for a mechanism that combines regulation and positive feedback to mediate precise temporal control of gene expression.

In its most basic form, timing can be achieved through the linear increase of a stable timekeeper molecule that induces a response when a specific abundance threshold is reached (*Ghusinga et al., 2017*). Using mathematical modeling, we have previously shown that the temporal precision at which this threshold is crossed is enhanced when the concentration of the timekeeper molecule increases in a non-linear fashion due to upstream regulation (*Gupta et al., 2018*). Moreover, we showed that such a non-linear increase can be mediated through either the gradual accumulation of a transcriptional activator or the gradual decrease of a transcriptional repressor, which induces the expression of the timekeeper gene in a threshold-dependent manner (*Gupta et al., 2018*). As an experimental model to test these predictions, we used the time-dependent expression of the Wnt receptor *mig-1* (*Mentink et al., 2014*; *Rella et al., 2021*). Quantification of *mig-1* expression during QR lineage progression showed that it is sharply upregulated in QR.pa, which is consistent with a non-linear mode of regulation. However, we found that *mig-1* is also expressed in the QR founder cell, raising the question whether these early and late phases of *mig-1* expression are linked. Examination of evolutionarily conserved cis-regulatory elements that we deleted from the endogenous *mig-1* locus using CRSPR/Cas9-mediated genome-editing showed that the two phases are independently regulated. Thus, we identified a small region in the first intron that is required for the early expression of *mig-1* in QR but is dispensable for the late phase of expression in QR.pa. Conversely, two partially redundant regions upstream of the *mig-1* coding sequence are required for the late phase of expression but have no detectable role in the early expression. The early phase of *mig-1* expression could therefore be disregarded in the analysis of the late phase of expression, which was confirmed using a dynamic model that was fit to the *mig-1* expression data.

The dynamic model also made predictions on the regulation of *mig-1* expression. In an activator-based model, removal of the activator disrupts the upregulation of *mig-1* expression in QR.pa, while in the case of a repressor, removal of the repressor results in premature expression of *mig-1* in QR.p. We found that *mig-1* expression in QR.pa was lost when the presumptive binding sites of the *mig-1* regulator were deleted from the *mig-1* upstream region, which supports the conclusion that the time-dependent expression of *mig-1* depends on transcriptional activation and not on repression.

Another prediction from our mathematical modeling is that the timing precision of a regulated timekeeper system can be enhanced when the timekeeper molecule also stimulates its own expression through positive feedback (*Gupta et al., 2020*). *mig-1* encodes a Frizzled-type Wnt receptor that triggers a canonical, BAR-1/β-catenin-dependent signaling pathway which induces the expression of target genes controlling QR.pa migration (*Mentink et al., 2014*; *Rella et al., 2021*). We found that the expression of *mig-1* is enhanced through BAR-1/β-catenin signaling, creating a positive feedback loop. Using the relative position at which QR.pa upregulates *mig-1* expression as a measure of timing precision, we discovered that timing was more variable when this feedback loop was disrupted through loss or constitutive activation of the BAR-1 pathway. Our modeling suggested that this effect is consistent with feedback on *mig-1* that acts both (1) independently of the activator and (2) through the activator pathway; either type alone was insufficient. We, therefore, speculate that BAR-1/β-catenin-dependent positive feedback on *mig-1* acts through multiple pathways. Our model assumed that disrupting the feedback disrupts the activator dynamics completely (the activator level becomes static in time). We also assumed that all of the positional variability in the disrupted cases comes from the expression dynamics and not, for example, from altered variability in the migration speed. Investigating feedback-independent contributions to the activator dynamics or perturbations to migration speed are interesting avenues for future study.

An intriguing aspect of the *mig-1* timer is that it acts across the division of QR.p into QR.pa and its apoptotic sister cell QR.pp. Inhibition of QR.p mitosis and cell cycle progression through depletion of the M-phase regulator CDK-1 showed that the temporal regulation of *mig-1* expression is independent of the division, ruling out a model in which the expression of *mig-1* is linked to the invariant developmental timing of *C. elegans* cell divisions. However, even though the division itself is not necessary for the upregulation of *mig-1* expression, we found that the asymmetry of the QR.p division influences the expression dynamics of *mig-1*. Our mathematical model allowed us to make predictions regarding the effect of cell division on the timing of *mig-1* expression upregulation. First,

the model assumed that upregulation is governed by the activator molecule number, not concentration. This assumption is justified by our finding that the activator acts transcriptionally in a cell whose nuclear size does not change upon division. However, for regulators that act post-transcriptionally or for cells whose nuclear volume changes upon division, the model would predict that the upregulation is governed by the cellular or nuclear concentration of the regulator, respectively, and not its molecule number. Second, the model assumes that the activator and *mig-1* transcripts are partitioned at division according to cell volume. This assumption is critical, particularly for the activator. If instead the activator were sequestered to QR.pa, for example, the molecule number would not change after division, and the dynamics of *mig-1* expression upregulation, therefore, would not depend on the QR.pa/QR.pp size ratio. Consequently, we predict that the activator is not sequestered at division and that unequal partitioning of the *mig-1* activator between the differently sized daughter cells is necessary for the upregulation of *mig-1* expression in QR.pa.

Examination of conserved cis-regulatory elements upstream of *mig-1* showed that two small, partially redundantly acting regions are required for the upregulation of *mig-1* expression in QR.pa. Chromatin immunoprecipitation followed by deep sequencing experiments (*Araya et al., 2014*) show that both regions overlap with binding peaks of multiple transcription factors, including homeobox transcription factors that are known to regulate Q neuroblast descendant migration (*Clark et al., 1993*), providing an entry point for the identification of the transcriptional regulators that control *mig-1* expression. Molecular insight into the expression dynamics and activity of these regulators will provide further insight into how robust timing precision can be mediated through a regulator-based timekeeping mechanism.

## Materials and methods

### Key resources table

| Reagent type (species) or resource | Designation | Source or reference | Identifiers | Additional information |
|---|---|---|---|---|
| Gene (*Caenorhabditis elegans*) | *mig-1* | Wormbase | Wormbase ID: WBGene00003238 | |
| Gene (*C. elegans*) | *cdk-1* | Wormbase | Wormbase ID: WBGene00000405 | |
| Gene (*C. elegans*) | *ced-3* | Wormbase | Wormbase ID: WBGene00000417 | |
| Gene (*C. elegans*) | *pig-1* | Wormbase | Wormbase ID: WBGene00021012 | |
| Gene (*C. elegans*) | *bar-1* | Wormbase | Wormbase ID: WBGene00000238 | |
| Strain and strain background (*C. elegans*) | For strains used in this study, see *Supplementary file 1b* | This paper | | Strains are available upon request. |
| Recombinant DNA reagent | *Pegl-17::tir1::TagBFP::unc-54* 3' UTR in pDESTR4-R3 | This paper | pKN618 | See Materials and methods. Plasmid is available upon request. |
| Sequence-based reagent | For sgRNA and ssODN sequences used for CRISPR/Cas9-based gene editing, see *Supplementary file 1c* | This paper | | |
| Chemical compound and drug | Auxin (natural indole-3-acetic acid) | Alfa Aesar | #A10556 | |
| Software and algorithm | R Studio | https://www.r-project.org/ | | |
| Software and algorithm | MATLAB | https://www.mathworks.com/ | | |

### *C. elegans* strains and culture

All *C. elegans* strains were grown at 20°C using standard culture conditions unless otherwise stated. The Bristol N2 strain was used as a wild type control. The alleles and transgenes generated and

used in this work are: LGI: *mig-1(hu295)*[Δintron A]; *mig-1(hu299)*[Δintron B]; *mig-1(hu314)*[Δupstream 1]; *mig-1(hu315)*[Δupstream 2]; *mig-1(hu335)*[Δupstream 1–2]; LGIII: *huIs210*[*Pegl-17::tir1::TagBFP, Pmyo-2::TdTomato*], *cdk-1(hu277*[*AID::cdk-1*]); *mab-5(gk670)*; LGIV: *huIs179*[*Pegl-17::ΔN-bar-1, Pmyo-2::mCherry*]; *pig-1(gm344)*; *ced-3(n717)*; LGV: *heIs63*[*Pwrt-2::GFP::PH, Pwrt-2::GFP::H2B, Plin-48::mCherry*]; LGX: *bar-1(ga80)*; *huIs166*[*Pwrt-2::mCherry::PH, Pwrt-2::mCherry::H2B, dpy-20(+)*]. Details on CRISPR/Cas9-mediated genome edits can be found in *Supplementary file 1a* and a list of strains used in this study is in *Supplementary file 1b*.

## *C. elegans* expression constructs and transgenesis

The Q lineage-specific TIR1 expressing transgene *huEx744* was generated by microinjection of expression construct pKN618 (*Pegl-17::tir1::TagBFP::unc-54* 3' untranslated region [UTR]) in the pDESTR4-R3 backbone plasmid at 20 ng/µl, using 5 ng/µl *Pmyo-2::mCherry* as co-injection marker. pBluescriptII was added for a final DNA concentration of 150 ng/µl in the microinjection mix. *huIs210* was generated by genomic integration of *huEx744* using gamma irradiation.

## CRISPR/Cas9-mediated genome editing

The allele *cdk-1(hu277*[*AID::cdk-1*]) was generated using CRISPR/Cas9-based genome editing, *dpy-10* co-conversion, and single-stranded DNA repair templates as previously described (*Paix et al., 2015*). Alleles *mig-1(hu295)* and *mig-1(hu299)* were generated using CRISPR/Cas9-based genome editing, *pha-1* co-conversion, and single-stranded DNA repair templates (*Ward, 2015*). Alleles *mig-1(hu314)*, *mig-1(hu315)*, and *mig-1(hu335)* were generated using CRISPR/Cas9-based genome editing as described (*Dokshin et al., 2018*). PCR fragments containing the T7 promoter and the single guide RNA (sgRNA) sequence of interest were used to synthesize sgRNA in vitro. Repair templates and sgRNAs were co-injected with recombinant SpCas9 (*D'Astolfo et al., 2015*). sgRNAs and repair template sequences can be found in *Supplementary file 1c*.

## Analysis of QR descendant migration

The final position of the QR descendant QR.pap was determined using widefield fluorescence microscopy at the late L1 larval stage (between 12 and 16 hr after hatching). In all experiments, transgenic animals expressing GFP in the seam cells and the Q lineage were used (transgene *heIs63*; *Wildwater et al., 2011*). The position of QR.pap was determined by measuring the distance of QR.pap to the seam cells V1.p and V2.p. This distance was then normalized to the distance between V1.p and V2.p to determine the relative position of QR.pap to these cells. The average of two position measurements (one using V1.p and one using V2.p) was used in the analyses. Measurements were performed using the FIJI package (*Schindelin et al., 2012*).

## Single-molecule FISH

The smFISH protocol was performed as described (*Ji et al., 2013*; *Ji and van Oudenaarden, 2012*). In all experiments, transgenic animals expressing GFP or mCherry in the seam cells and the Q neuroblast lineage were used (*heIs63* and *huIs166*, respectively; *Wildwater et al., 2011*). Synchronized L1 larvae were fixed using 4% paraformaldehyde and kept in 70% ethanol. Hybridization was performed overnight at 37°C, in the dark. Oligonucleotide probes used in hybridization were designed using the Stellaris probe designer and chemically coupled to fluorescent dyes Cy5 (*mig-1*; *Ji and van Oudenaarden, 2012*) or ATTO565 (*larp-1*, a positive control for smFISH staining quality). Animals were incubated in buffer containing DAPI for nuclear counterstaining prior to mounting. Z-stacks of 0.2–0.3 µm thickness were collected on a Perkin Elmer Ultraview VoX mounted on a Leica DMI6000 microscope with a 100×/NA 1.47 oil objective, or a Nikon Ti2 microscope with a 100×/NA 1.45 oil objective. Both systems used a Yokogawa X1 spinning disk and Hamamatsu Orca Flash4.0 V3 camera. Images were acquired at 1024×1024 resolution. mRNA spots were quantified manually from the z-stacks. Only spots detected in two or more consecutive focal planes were counted. Image analysis was performed in FIJI (*Schindelin et al., 2012*).

## AID of CDK-1

The *Arabidopsis thaliana* TIR1 protein was expressed as an integrated transgene driven by the *egl-17* promoter (*huIs210*; *Branda and Stern, 2000*). This ensured CDK-1 was only depleted in the Q

neuroblast lineage. By transferring L1 larvae to plates containing 0.5 mM natural auxin indole-3-acetic acid (Alfa Aesar, #A10556) during the time window between QR and QR.p division (300–345 min after hatching), QR.p division was specifically blocked.

## Quantification of DNA content

DNA content was determined using images from smFISH experiments as follows. Per cell, the range of focal planes spanning the nucleus was determined. A z-stack of the focal planes with the nucleus was then made using the 'sum slices' option in FIJI. A region of interest (ROI) was subsequently drawn around the nucleus, and total intensity was measured in the ROI. Intensity measurements from multiple seam (Vn.p) cells were averaged as the 2C internal control for each animal. The intensity of DAPI staining in QR.p was then divided by this average internal control to produce the DNA content ratio. A ratio of 1 corresponds to 2C, a ratio of 2–4C, and a ratio of 4–8C.

## Identification of evolutionarily conserved regions in the *mig-1* upstream region and first intron

The homologous gene of *C. elegans mig-1* was first identified in 22 other *Caenorhabditis* species using the BLASTP algorithm on the following genomes: *Caenorhabditis monodelphis* (JU1667_v1), *Caenorhabditis plicata* (SB355_v1), *Caenorhabditis virilis* (JU1968_v1), *Caenorhabditis quiockensis* (C. sp. 38, JU2809_v1), *Caenorhabditis castelli* (JU1956_v1), *Caenorhabditis uteleia* (C. sp. 31, JU2585_ v1), *Caenorhabditis afra* (JU1286_v1), *Caenorhabditis sulstoni* (C. sp.32, JU2788_v1), *Caenorhabditis waitukubuli* (C. sp. 39, NIC564_v1), *Caenorhabditis panamensis* (C. sp. 28, QG2080_v1), *Caenorhabditis becei* (C. sp. 29, QG2083_v1), *Caenorhabditis nouraguensis* (JU2079_v1), *Caenorhabditis kamaaina* (QG2077_v1), *Caenorhabditis inopinata* (C. sp. 34, NK74SC_v1), *Caenorhabditis tropicalis* (JU1373_301), *Caenorhabditis doughertyi* (JU1771_v1), *Caenorhabditis brenneri* (C_brenneri-6.0.1b), *Caenorhabditis latens* (PX534_v1), *Caenorhabditis briggsae* (CB4), *Caenorhabditis sinica* (JU800_v1), *Caenorhabditis tribulationis* (C. sp. 40, JU2818_v1), and *Caenorhabditis zanzibari* (C. sp. 26, JU2190_ v1). BLASTP was performed on this set of species on http://caenorhabditis.org/. We then looked for motif conservation in *mig-1* regulatory regions in this set of species using the MEME Suite tools (*Bailey et al., 2009*) with the following parameter settings: -mod zoops, -nmotifs 4, -minw 8, and -maxw 18 for the first intron and -mod zoops, -nmotifs 4, -minw 10, and -maxw 25 for the upstream region.

## Fitting of mathematical model to data

We find the parameters of either the increasing activator model or the decreasing repressor model (*Figure 2D*) using a least-squares fit to the control data (*Figure 2B*). Specifically, to find $m\left(0\right)$ and $\nu$, we perform a linear fit to the log of the *mig-1* mRNA number for the QR data only, corresponding to exponential decay; we find $m\left(0\right) = 31$ mRNAs and $\nu^{-1} = 0.65$ (in the arbitrary time units of *Figure 2D*). Then, to find the remaining parameters, we fit the numerical solution of the *dm/dt* equation to all of the control data, minimizing the sum of the squares of both the vertical and horizontal residuals, scaled by the average value of mRNA number and time across all data, respectively. For the activator model, we find $\alpha = 390$, $K/k = 3.0$, and $H = 14$. For the repressor model, we find $\alpha = 130$, $\mu H = 6.7$, and $Hln\left(r_0/K\right) = 18$. The high value of the Hill coefficients here and below is a reflection of the fact that *mig-1* expression remains low in QR.p cells and then increases very sharply in QR. pa cells. Because we model this increase with a single regulating species, whereas *mig-1* is likely regulated by more than one species in reality, the Hill coefficient can be viewed as an effective parameter rather than a biochemical constant. Furthermore, we see that some parameters need only be determined in particular combinations. The reason is the following. For the activator model, plugging $a\left(t\right) = kt$ into $F\left(t\right) = \alpha a^H/\left(a^H + K^H\right)$ obtains $F\left(t\right) = \alpha/\left[1 + \left(K/kt\right)^H\right]$, which depends only on the ratio $K/k$. For the repressor model, plugging $r\left(t\right) = r_0 e^{-\mu t}$ into $F\left(t\right) = \alpha K^H/\left(r^H + K^H\right)$ obtains $F\left(t\right) = \alpha/\left[1 + \left(r_0 e^{-\mu t}/K\right)^H\right] = \alpha/\left\{1 + exp\left[Hln\left(r_0/K\right) - \mu Ht\right]\right\}$, which depends only on the combinations $Hln\left(r_0/K\right)$ and $\mu H$. Although these parameter combinations are sufficient to specify the *mig-1* dynamics, for illustrative purposes, we plot example activator and repressor dynamics consistent with these values in *Figure 2D* by choosing $K = 20$ for the activator model (giving $k = 6.6$), and $K = 5$

and $H = 14$ for the repressor model (giving $r_0 = 19$ and $\mu = 0.48$). For the zero initial *mig-1* condition (**Figure 2D**), we set $m(0) = 0$.

To find the division time from the control data (**Figure 4A**), we count the cumulative number of QR.p data points as a function of position (moving posteriorly), count the cumulative number of QR.pa data points as a function of position (moving anteriorly), and find the position at which these numbers are equal. This position is $T = 2.32$ in the arbitrary time units of **Figure 4E**. The QR.pa dynamics from the model with division, using the wild type mean size ratio $\rho = 3.5$, are fit to the control data using the same procedure as above, yielding the parameters $\alpha = 240$, $K/k = 2.6$, and $H = 17$. This fit is shown for QR.p, QR.pa, and QR.pp in **Figure 4E** (dark colors). To illustrate the effect of decreasing size ratio $\rho$, we also plot these dynamics with $\rho = [2.7, 1.8, 1]$ in **Figure 4E** (light colors). To illustrate the dependence of the QR.pa *mig-1* molecule number $m(t)$ on the size ratio $\rho$, we plot $m(t)$ vs. $\rho$ at times between $t = 3$ and $t = 3.3$ (gray region) and at the midpoint time $t = 3.15$ (black line) in **Figure 4F**.

## Modeling the effect of feedback on timing precision

To model the effect of BAR-1/$\beta$-catenin dependent feedback on the noise in *mig-1* expression, we consider three possibilities: (1) the feedback acts on *mig-1* independently of the activator (**Figure 5—figure supplement 2A**, left), (2) the feedback acts on *mig-1* through the activator (**Figure 5—figure supplement 2A**, middle), or (3) both (**Figure 5—figure supplement 2A**, right). In all cases, we neglect cell division and *mig-1* degradation for simplicity, and we set $t = 0$ using the rightmost position data point in **Figure 5A** (ΔN-BAR-1). We calculate the mean $\bar{t}$ and variance $\sigma_t^2$ in the time at which $m$ reaches its threshold $m_*$ numerically from the master equation following our previous work (**Gupta et al., 2018**).

Possibility 1 (**Figure 5—figure supplement 2A**, left) corresponds to the dynamics $a(t) = kt$ and $dm/dt = F(a, m)$. Both the *bar-1(ga80)* null mutant and the constitutively active ΔN-BAR-1 break the feedback, corresponding to $F(a, m) = F(a)$. In these cases, we take $F(a) = \alpha/\left[1 + (K/a)^H\right]$ as before. For the control case, previous work **Gupta et al., 2020** found that this topology reduces noise when the feedback is negative for low $a$ and positive for high $a$. The simplest modification that achieves this property is $F(a, m) = \alpha/\left[1 + (K/a)^{H_0 m}\right]$, corresponding to a regulation function that sharpens with $m$. Because breaking the feedback can also affect the activation parameters for this topology, we fit the *bar-1(ga80)* and ΔN-BAR-1 cases separately from the control case. Fitting as above results in $\alpha = 260$, $K/k = 3.4$, and $H_0 = 6.7$ for the control (**Figure 5—figure supplement 2B**, left, red); $\alpha = 490$, $K/k = 4.2$, and $H = 13$ for *bar-1(ga80)* (**Figure 5—figure supplement 2B**, middle, red); and $\alpha = 490$, $K/k = 4.2$, and $H = 6$ for ΔN-BAR-1 (**Figure 5—figure supplement 2B**, right, red). Calculation of $\bar{t}$ and $\sigma_t^2$ requires specifying $k$, and we find that for no $k$ value is the $CV = \sigma_t/\bar{t}$ significantly increased upon breaking the feedback (**Figure 5—figure supplement 2C**, middle; **Figure 5—figure supplement 2** shows the results for $k = 30$ and $m_* = 25$.

Possibilities 2 (**Figure 5—figure supplement 2A**, middle) and 3 (**Figure 5—figure supplement 2A**, right) both correspond to the dynamics $da/dt = G(m)$, but possibility 2 has $dm/dt = F(a)$, whereas possibility 3 has $dm/dt = F(a, m)$. Because we found above that breaking autoregulatory feedback has little effect on the $CV$, we simplify the dynamics in possibility 3 to also read $dm/dt = F(a)$. Nevertheless, we retain the key topological difference that breaking the feedback cannot change the activation parameters in possibility 2 but can in possibility 3. Thus, in possibility 2, the *bar-1(ga80)* and ΔN-BAR-1 cases inherit the activation parameters from the control case, whereas in possibility 3, the *bar-1(ga80)* and ΔN-BAR-1 cases are fit separately from the control case.

For the control case in possibilities 2 and 3, we take $F(a) = \alpha_0 + \alpha/\left[1 + (K/a)^H\right]$ and analogously $G(m) = \beta_0 + \beta/\left[1 + (J/m)^H\right]$, where the Hill coefficients are equal for simplicity, and at least one of $\alpha_0$ and $\beta_0$ must be nonzero to initiate production from zero molecules. Previous work **Gupta et al., 2020** found that this topology reduces noise when the times at which $a$ and $m$ cross their thresholds $K$ and $J$, respectively, are well separated. We focus on the regime where $a$ crosses first ($K \ll J$), although we will see below that noise is reduced even for $K > J$. To reduce the number of fitting parameters, we recognize that the limit $K \ll J$ corresponds to the simplification $G(m) = \beta_0$. Therefore we fit in this limit, with $\alpha_0 = 0$, and we later check the robustness of our results as we reintroduce $J$ and $\beta$. Fitting results in $\alpha = 2800$, $K/\beta_0 = 4.5$, and $K/k = 4.2$ (**Figure 5—figure supplement 2B**, left, cyan and purple).

For the *bar-1(ga80)* and ΔN-BAR-1 cases in possibilities 2 and 3, we recognize that, unlike in possibility 1, breaking the feedback disrupts the production of the activator. In these cases, we take $a$ to be a constant $a_0$, and therefore, we have $dm/dt = F(a_0)$. As discussed above, in possibility 2, the parameters of $F$ are inherited from the control case. Consequently, the *bar-1(ga80)* case, which corresponds to a low value of $a_0$, produces a shallow linear increase in *mig-1* (***Figure 5—figure supplement 2B***, middle, cyan); while the ΔN-BAR-1 case, which corresponds to a high value of $a_0$, produces a steep linear increase in *mig-1* (***Figure 5—figure supplement 2B***, right, cyan). Because neither is a reasonable description of the data, we conclude that possibility 2 is not a good model of the feedback, and we do not consider the *CV* for possibility 2.

In possibility 3, as discussed above, the *bar-1(ga80)* and ΔN-BAR-1 cases are fit separately from the control case. Fitting results in $F(a_0) = 5.1$ for *bar-1(ga80)* (***Figure 5—figure supplement 2B***, middle, purple) and $F(a_0) = 7.7$ for ΔN-BAR-1 (***Figure 5—figure supplement 2B***, right, purple). The *CV* in these cases can be calculated analytically (***Gupta et al., 2018***) as $CV = m_*^{-1/2}$. The control case requires specifying $\beta_0$, and we find that for $\beta_0 \gtrsim 10$, the *CV* is increased upon breaking the feedback (***Figure 5—figure supplement 2C***, right) in agreement with the experimental data (***Figure 5B*** and ***Figure 5—figure supplement 2C***, left). Indeed, we find that this property remains true even for $J$ as small as 1 and for $\beta \gg \beta_0$. ***Figure 5B*** (inset) and ***Figure 5—figure supplement 2*** show the results for $\beta_0 = 25$, $J = 10$, $\beta = 75$, and $m = 25$.

### Statistical ANOVA data

To compare the CVs of *mig-1* expression in *bar-1* mutants (***Figure 5B***), we performed a Brown–Forsythe ANOVA test. Specifically, we made two comparisons (control vs. *bar-1* null and control vs. ΔN-BAR-1) and, therefore, multiplied p-values by two to account for the Bonferroni correction. Position data were included if the *mig-1* mRNA molecule number was greater than or equal to a threshold and rescaled by their mean to obtain the CV. We performed the test for thresholds in the range 10–25. For the control vs. *bar-1* null comparison, the Bonferroni-corrected p-values ranged from 0.07 to 0.9, with a mean value of 0.4. For the control vs. ΔN-BAR-1comparison, the Bonferroni-corrected p-values ranged from $3 \times 10^{-6}$ to 0.008, with a mean value of 0.001.

### Statistical testing and modeling software

Statistical testing and plot generation was performed using R 4.1.0 and RStudio 1.4.1717, using various packages from the tidyverse. Mathematical modeling was performed in Matlab.

### Data availability statement

Source datafiles containing the numerical data used to generate the figures can be accessed at https://github.com/erikschild/mig1_timer_code, (copy archived at ***Schild, 2023a***) and the code used for modeling can be found at https://github.com/amugler/mig1, (copy archived at ***Schild, 2023b***).

## Acknowledgements

We thank the Hubrecht Imaging Center for technical support. This work was funded by the Human Frontier Science Program Grant RGP0030/2016 to AM, MAF, and HCK and grants from the Simons Foundation (376198) and the National Science Foundation (PHY-1945018) to SG and AM. Some strains were provided by the CGC, which is funded by the NIH Office of Research Infrastructure Programs (P40 OD010440). We are grateful to Damien Coudreuse and members of the Korswagen and Galli groups for helpful discussions and critical reading of the manuscript.

## Additional information

### Funding

| Funder | Grant reference number | Author |
|---|---|---|
| Human Frontier Science Program | RGP0030/2016 | Marie-Anne Félix<br>Andrew Mugler<br>Hendrik C Korswagen |
| National Science Foundation | PHY-1945018 | Shivam Gupta<br>Andrew Mugler |
| Simons Foundation | 376198 | Shivam Gupta<br>Andrew Mugler |

The funders had no role in study design, data collection and interpretation, or the decision to submit the work for publication.

### Author contributions

Erik S Schild, Conceptualization, Data curation, Software, Formal analysis, Methodology, Writing – original draft, Writing – review and editing; Shivam Gupta, Conceptualization, Investigation; Clément Dubois, Investigation, Methodology; Euclides E Fernandes Póvoa, Investigation, Writing – review and editing; Marie-Anne Félix, Conceptualization, Supervision, Funding acquisition, Project administration, Writing – review and editing; Andrew Mugler, Conceptualization, Formal analysis, Supervision, Funding acquisition, Project administration, Writing – review and editing; Hendrik C Korswagen, Conceptualization, Data curation, Supervision, Funding acquisition, Writing – original draft, Project administration, Writing – review and editing

### Author ORCIDs

Euclides E Fernandes Póvoa http://orcid.org/0000-0002-0920-7783
Andrew Mugler http://orcid.org/0000-0001-9367-7026
Hendrik C Korswagen http://orcid.org/0000-0001-7931-4472

### Decision letter and Author response

Decision letter https://doi.org/10.7554/eLife.82675.sa1
Author response https://doi.org/10.7554/eLife.82675.sa2

## Additional files

### Supplementary files

• Supplementary file 1. Details on gene edits, strains and sequence based reagents. **a**. CRISPR/Cas9-mediated gene edits of *mig-1* and *cdk-1*. **b**. *C. elegans* strains used in this study. **c**. Single guide RNA (sgRNA) and single-stranded oligodeoxynucleotide (ssODN) sequences used for CRISPR/Cas9-mediated gene editing.

• MDAR checklist

### Data availability

All data generated or analysed during this study are included in the manuscript and figures. Source datafiles containing the numerical data used to generate the figures can be accessed at https://github.com/erikschild/mig1_timer_code, (copy archived at *Schild, 2023a*).

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
