## [Editor Report]

This paper deals with an important unsolved problem in developmental biology: how cells execute their dynamics at the right time. The study combines compelling quantitative single-cell and single-transcript experiments with genetic perturbations and computational modelling and provides important insights into how the timing of transcription is regulated.

---

## [Decision Letter]

**Decision letter after peer review:**

Thank you for submitting your article "Precise temporal control of neuroblast migration through combined regulation and feedback of a Wnt receptor" for consideration by *eLife*. Your article has been reviewed by 2 peer reviewers, and the evaluation has been overseen by a Reviewing Editor and Marianne Bronner as the Senior Editor. The reviewers have opted to remain anonymous.

Essential revisions:

1. Both reviewers raise the lack of integration between the model and data.

2. Secondly, the evidence for the positive feedback loop is weak/lacking. Additional experiments would be optimal. Alternatively, modelling could be used to interpret or strengthen these results, but this would require explicitly including the positive feedback loop and stochastic variability. At least it would be important to substantially rephrase the conclusions, to recognize that the evidence is relatively weak. I refer you to the comments of the reviewers for details.

*Reviewer #1 (Recommendations for the authors):*

- l. 151. "2C". Given the article's mixed audience of biologists and modelers, it would be good to explain the 2C, 4C, and 8C notation. Currently, it is not easy to deduce the meaning of 2C for the article text, if you do not already know it.

- l. 154. It is not explained why it is expected that CDK1-inhibited neuroblasts are 4C, not 2C. This is presumably because inhibiting CDK1 does not inhibit DNA duplication but does inhibit the subsequent division. It would be good to mention this explicitly.

- l. 189-208. In general, the fitting procedures outlined in this section and the Materials and methods seem fine, but it would be helpful if they would be presented in a way that allows visual comparison of the quality of the fit. Currently, smFISH data is always shown separately from fits, and plots with fitting data are mostly used to highlight predictions of the model. It would be good if somewhere, either in the main text or in the SI, the fitting data is plotted on top of the data, just to see how good the fit actually is. I realize that data and fit have different time axes, but this should not be a fundamental problem in plotting data and fit together if I understand the fitting procedure correctly.

- l. 189-208, l. 508-533. The fitted parameter values are only mentioned in the Methods. The values for the Hill coefficients for both the repressor and activator model are very high, H=14 or H=17, but this is not really commented on. I understand from the rest of the paper that mig-1 likely has some form of positive feedback on its own expression, which could in principle explain high Hill coefficients. But are such high Hill coefficients consistent with such a positive feedback mechanism? I think the authors should discuss the high Hill coefficient and whether such high values are plausible based on their proposed mechanism.

- l. 228, Figure 3A, C. The model predicts that decreasing the activator (hereby mutating activator binding sites) leads to a slower increase in mig-1 level. In the experiments, this is not visible for Δ upstream 1, but here a substantial number of animals appear to fail in inducing mig-1. For Δ upstream 2, there are even more animals like this. This fact is mentioned in the main text, but not interpreted in any way. How do the authors interpret this in light of the activator model?

- l. 245. 'required for the rapid upregulation'. To me, this formulation seems too strong. Looking at Figure 4A, I still see rapid upregulation in pig-1 mutants, as in happening quickly after the division of QR.p, but with lower mig-1 levels reached compared to the control.

- l. 245-298, Figure 4. Related to the above point, it is not clear to me how the authors interpret the pig-1 experiments in relation to the fitting results in Figure 4E. The model predicts that decreasing the size ratio between QR.pa and QR.pp leads to a slower accumulation of mig-1, but with the same mig-1 level ultimately reached, albeit at a later time. The pig-1 data in Figure 4A could be consistent with slower accumulation, but, in contrast to Figure 4E, mig-1 expression never reaches the peak control level. How do the authors interpret this contrasting result? An alternative model to explain the data could be that in pig-1 mutants mig-1 expression is induced at the same time as in control animals, but with a lower 'amplitude' of mig-1 expression. Can the authors rule out such an alternative model? Finally, in the author's model QR.pa migration stops once a specific mig-1 level is reached. In that case, wouldn't one expect a positional defect for pig-1 mutants? I didn't see this mentioned anywhere in the article.

- l. 281-285. I found the explanation here confusing. I think I understand the main point: the cells may have different sizes, but have the same nuclear volume. As the activator is likely in the nucleus, one can therefore just use the same molecular level as before, without taking into account the difference in volume. However, the way it is written here suggested initially to me that a modification was made to the model by focusing on a number of molecules rather than concentration, even though, if I understand correctly, the model was formulated in terms of a number of molecules from the beginning. It would be good to clarify this.

- l. 296-298. I have the same issue here as above: it is not clear to me that division asymmetry is required for rapid upregulation. The proposed role of asymmetry can also be formulated more sharply. If I understand it correctly, according to the model, the only impact of size asymmetry is in the asymmetric inheritance of the activator. In particular, the subsequent rate of production of the activator in Q.pa does not depend on its size, correct? Or am I missing something?

- l. 300-329. I find this section the weakest of the manuscript, both in terms of experiments and modelling. The main aim of this section is to test the model prediction from an earlier paper (Gupta 2020) that positive feedback of mig-1 on its own expression reduces variability in timing. The main experimental weakness is that the authors do not specifically perturb this positive feedback loop, but instead use two Wnt mutants (the loss-of-function bar-1(0) and constitutively active DeltaN-BAR-1) that likely impact many genes, including those responsible for migration (l. 95,96), as also evidenced by the resulting change in average Q.pa position. While the authors find an increase in variability of Q.pa position, this could also be explained by changes in the expression of any of the other BAR-1 targets. The proper experiment, I think, would be to remove the POP-1/TCF sites in the mig-1 promoter, although there might be experimental reasons why this is not feasible. The last sentence of the section (l. 328-329) is a bit careful in drawing strong conclusions from these result, but this caution is absent from the abstract and introduction. I also think that the limits of the BAR-1 experiments should be explicitly discussed.

In terms of modelling, I found it surprising that, in contrast to the rest of the paper, here the experiments are not compared to the model (apart from a brief reference to the main conclusion of an earlier publication). It seems to me that the experimental results raise questions that are not addressed by these earlier publications. For instance, DeltaN-BAR-1 animals show no increase in mig-1 levels, indicating that Wnt signaling is completely saturated. Would the model predict an increase in variability in timing (and position) when Wnt signaling is saturated? That seems to me, at first sight, a different kind of perturbation than removing positive feedback, e.g. by removing mig-1's POP-1/TCF sites. I would think that for saturated Wnt signaling, mig-1 expression becomes less dependent on the activator level, and thus less susceptible to variability in the activator level.

- l. 405-412. Unless there are very many transcription factors that bind these sequences, it seems a bit odd to not mention the identity of these transcription factors.

- l. 516-517. Apparently, some parameters, like H, r0, and K, are only constrained by the experimental data in specific ratios. It would be good if the authors provide some form of explanation for this.

-l. 526. "T=2.32". What does this time correspond to in terms of cell position?

- Figure 1D. What is the GFP reporter?

- Figure 2-supplement 2. What statistical test has been used? I am also asking because from looking at the data it is not so clear that control and Δ intron A are really not significantly different. What is the P value?

- Figure 4E. The figure shows that decreasing the size ratio leads to slower mig-1 induction, but it is not clear how well it fits the data. If you split the data in Figure 4C into brackets of different ratios/rho, would it then fit the model data for the rho of that bracket? Based on Figure 4E, F, it remains open to me how well the model actually fits the data.

- l.823-824. What is this a test of? I am confused because the text mentions transcripts, but the panel is about position.

*Reviewer #2 (Recommendations for the authors):*

The strength of the manuscript is the overall question, the experimental data, and the simple model. But given the quality of the data, the data analysis, model fitting, and predictions could be improved or at least better presented.

Specifically, I would like to see how much variation exists between different replica experiments. Right now, all the RNA-FISH data is pooled and not separated by replica experiments. One simple way to address this would be to plot the data points separated by different symbols specific to each biological replica experiment. This comment is relevant to all figures with data.

How did the authors determine a threshold in these data sets? Applying a piecewise linear fit to the experimental data without the model enables very precise quantification of the threshold. This should be done for each replica experiment to determine the mean and standard deviation in the threshold. This should be also done for all the WT and mutant data sets to show how the expression levels and threshold are effects by different mutants.

With regard to the model, the authors need to show how the model fits to the data, by overlaying the model and the data in the same plot. The authors state in the manuscript that the spatial position of the cells is proportional to time, so this should be possible. This should be done for each replica to determine the mean and standard deviation in the model parameter. The model parameters and their errors should be added to the manuscript.

By comparing the model fits and predictions using the estimated parameter errors, authors can better determine if the differences seen in the WT and mutant experiment are significantly large to explain this difference by changing specific rates in the model.

Although the authors quantify the position equals temporal variance in the mig1 expression, they do not use their model to make this prediction or even analyze the data with their model. Why is this? Can the model not compute temporal variances? Please clarify.

---

## [Author Response]

Essential revisions:1. Both reviewers raise the lack of integration between the model and data.

We now provide additional supplemental figures in which the quantitative gene expression data is superimposed on the different model fits.

2. Secondly, the evidence for the positive feedback loop is weak/lacking. Additional experiments would be optimal. Alternatively, modelling could be used to interpret or strengthen these results, but this would require explicitly including the positive feedback loop and stochastic variability. At least it would be important to substantially rephrase the conclusions, to recognize that the evidence is relatively weak. I refer you to the comments of the reviewers for details.

We now explicitly model the positive feedback and compare our findings to the experimental data. We find that when *mig-1* is subjected to feedback, both through the activator and independently from the activator, then disrupting the feedback either through loss or constitutive activation leads to increased variability in the timing of *mig-1* expression, which is consistent with the experiments.

Reviewer #1 (Recommendations for the authors):- l. 151. "2C". Given the article's mixed audience of biologists and modelers, it would be good to explain the 2C, 4C, and 8C notation. Currently, it is not easy to deduce the meaning of 2C for the article text, if you do not already know it.

We have rewritten this part to explain in more detail the terminology and implications of the differences in DNA content.

- l. 154. It is not explained why it is expected that CDK1-inhibited neuroblasts are 4C, not 2C. This is presumably because inhibiting CDK1 does not inhibit DNA duplication but does inhibit the subsequent division. It would be good to mention this explicitly.

Yes, that is correct. We now explain this point more explicitly.

- l. 189-208. In general, the fitting procedures outlined in this section and the Materials and methods seem fine, but it would be helpful if they would be presented in a way that allows visual comparison of the quality of the fit. Currently, smFISH data is always shown separately from fits, and plots with fitting data are mostly used to highlight predictions of the model. It would be good if somewhere, either in the main text or in the SI, the fitting data is plotted on top of the data, just to see how good the fit actually is. I realize that data and fit have different time axes, but this should not be a fundamental problem in plotting data and fit together if I understand the fitting procedure correctly.

We have added supplemental figures showing combinations of the data and corresponding fitting (see Figure 2, figure supplement 3 and Figure 4, figure supplement 1).

- l. 189-208, l. 508-533. The fitted parameter values are only mentioned in the Methods. The values for the Hill coefficients for both the repressor and activator model are very high, H=14 or H=17, but this is not really commented on. I understand from the rest of the paper that mig-1 likely has some form of positive feedback on its own expression, which could in principle explain high Hill coefficients. But are such high Hill coefficients consistent with such a positive feedback mechanism? I think the authors should discuss the high Hill coefficient and whether such high values are plausible based on their proposed mechanism.

The high value of the Hill coefficient is a reflection of the fact that *mig-1* expression remains low in QR.p cells, then increases very sharply in QR.pa cells. Because we model this increase with a single activating species, whereas *mig-1* is likely regulated by more than one species in reality, the Hill coefficient can be viewed as an effective parameter rather than a biochemical constant. The reviewer is correct that when we include feedback of *mig-1* on its own expression, as discussed below, we find that the Hill coefficient is lowered (from 14 or 17, to 12).

- l. 228, Figure 3A, C. The model predicts that decreasing the activator (hereby mutating activator binding sites) leads to a slower increase in mig-1 level. In the experiments, this is not visible for Δ upstream 1, but here a substantial number of animals appear to fail in inducing mig-1. For Δ upstream 2, there are even more animals like this. This fact is mentioned in the main text, but not interpreted in any way. How do the authors interpret this in light of the activator model?

The expression of *mig-1* is dependent on two, partially redundant cis-regulatory elements in the upstream region of the gene. These two elements may act cooperatively in *mig-1* activation, and therefore deletion of the individual elements may increase the threshold at which *mig-1* expression is induced by the activator. The small population of QR.pa cells that fail to upregulate *mig-1* expression in upstream deletion #1 and the larger population in upstream deletion #2 may represent cells in which this threshold is not reached within the time-window in which the cells were examined.

- l. 245. 'required for the rapid upregulation'. To me, this formulation seems too strong. Looking at Figure 4A, I still see rapid upregulation in pig-1 mutants, as in happening quickly after the division of QR.p, but with lower mig-1 levels reached compared to the control.

We agree and have adapted the text.

- l. 245-298, Figure 4. Related to the above point, it is not clear to me how the authors interpret the pig-1 experiments in relation to the fitting results in Figure 4E. The model predicts that decreasing the size ratio between QR.pa and QR.pp leads to a slower accumulation of mig-1, but with the same mig-1 level ultimately reached, albeit at a later time. The pig-1 data in Figure 4A could be consistent with slower accumulation, but, in contrast to Figure 4E, mig-1 expression never reaches the peak control level. How do the authors interpret this contrasting result? An alternative model to explain the data could be that in pig-1 mutants mig-1 expression is induced at the same time as in control animals, but with a lower 'amplitude' of mig-1 expression. Can the authors rule out such an alternative model? Finally, in the author's model QR.pa migration stops once a specific mig-1 level is reached. In that case, wouldn't one expect a positional defect for pig-1 mutants? I didn't see this mentioned anywhere in the article.

Our modeling suggests that a decrease in cell size ratio will lower the amount of activator that is segregated into QR.pa, and thereby delay the activation of *mig-1* expression. Since we measure in the same time-window as in the control animals, we see this effect as a reduction in expression. We would need measurements at later time-points to confirm that *mig-1* expression will further increase as predicted by the model, but this is not possible because QR.pa divides shortly after our analysis time-window.

The alternative model raised by the reviewer is that *pig-1* may directly influence the expression of *mig-1*. Although we cannot formally exclude this possibility, our observation that *mig-1* expression correlates with the degree of cell size asymmetry in *pig-1* mutants strongly supports our model that segregation of the activator, and not *pig-1* itself, determines the observed effects on *mig-1* expression.

As to the question on QR.pa position in *pig-1* mutants, we did not examine this because the reduction in QR.pa cell size may affect its ability to migrate, making these results difficult to interpret.

- l. 281-285. I found the explanation here confusing. I think I understand the main point: the cells may have different sizes, but have the same nuclear volume. As the activator is likely in the nucleus, one can therefore just use the same molecular level as before, without taking into account the difference in volume. However, the way it is written here suggested initially to me that a modification was made to the model by focusing on a number of molecules rather than concentration, even though, if I understand correctly, the model was formulated in terms of a number of molecules from the beginning. It would be good to clarify this.

Yes, all the modeling uses molecule number, not concentration. This point has been added to the main text.

- l. 296-298. I have the same issue here as above: it is not clear to me that division asymmetry is required for rapid upregulation. The proposed role of asymmetry can also be formulated more sharply. If I understand it correctly, according to the model, the only impact of size asymmetry is in the asymmetric inheritance of the activator. In particular, the subsequent rate of production of the activator in Q.pa does not depend on its size, correct? Or am I missing something?

We agree and have adapted the text.

- l. 300-329. I find this section the weakest of the manuscript, both in terms of experiments and modelling. The main aim of this section is to test the model prediction from an earlier paper (Gupta 2020) that positive feedback of mig-1 on its own expression reduces variability in timing. The main experimental weakness is that the authors do not specifically perturb this positive feedback loop, but instead use two Wnt mutants (the loss-of-function bar-1(0) and constitutively active DeltaN-BAR-1) that likely impact many genes, including those responsible for migration (l. 95,96), as also evidenced by the resulting change in average Q.pa position. While the authors find an increase in variability of Q.pa position, this could also be explained by changes in the expression of any of the other BAR-1 targets. The proper experiment, I think, would be to remove the POP-1/TCF sites in the mig-1 promoter, although there might be experimental reasons why this is not feasible. The last sentence of the section (l. 328-329) is a bit careful in drawing strong conclusions from these result, but this caution is absent from the abstract and introduction. I also think that the limits of the BAR-1 experiments should be explicitly discussed.

We agree that mutating POP-1/TCF binding sites in the *mig-1* promoter would more specifically address this question. Unfortunately, there are 12 predicted POP-1/TCF binding sites within the first 1 kb of upstream sequence, making it challenging to mutate all of these sites in the endogenous locus.

In terms of modelling, I found it surprising that, in contrast to the rest of the paper, here the experiments are not compared to the model (apart from a brief reference to the main conclusion of an earlier publication). It seems to me that the experimental results raise questions that are not addressed by these earlier publications. For instance, DeltaN-BAR-1 animals show no increase in mig-1 levels, indicating that Wnt signaling is completely saturated. Would the model predict an increase in variability in timing (and position) when Wnt signaling is saturated? That seems to me, at first sight, a different kind of perturbation than removing positive feedback, e.g. by removing mig-1's POP-1/TCF sites. I would think that for saturated Wnt signaling, mig-1 expression becomes less dependent on the activator level, and thus less susceptible to variability in the activator level.

We now explicitly model the positive feedback and compare our findings to the experiments. Specifically, we find that when *mig-1* is subjected to feedback, both through the activator and independently from the activator, then disrupting the feedback either through loss or constitutive activation leads to increased variability in the timing of *mig-1* expression, consistent with the experiments (see the new Figure 5B, Figure 5, figure supplement 2, and Materials and methods section “Modeling the effect of feedback on timing precision”).

These new results account for saturation of Wnt signaling in ΔN-BAR-1 animals by making the activator saturated at a constant, time-independent level in this case. The *mig-1* expression level then indeed becomes effectively independent of the activator, and thus not subject to variability in the activator level. However, it also loses the benefit of the activator dynamics to create the sharp increase in *mig-1* expression, which acts to reduce variability. The net result, the modeling shows, is that the *mig-1* dynamics become more linear and more variable, consistent with the experiments.

- l. 405-412. Unless there are very many transcription factors that bind these sequences, it seems a bit odd to not mention the identity of these transcription factors.

Transcription factors that bind to these regions include homeobox transcription factors such as LIN-39 and MAB-5 that regulate Q neuroblast descendant migration. This has been added to the main text.

- l. 516-517. Apparently, some parameters, like H, r0, and K, are only constrained by the experimental data in specific ratios. It would be good if the authors provide some form of explanation for this.

For the activator model, plugging a(t)=kt into F(t)=αaH/(aH+KH)obtains F(t)=α/[1+(K/kt)H], which depends only on the ratio K/k. For the repressor model, plugging r(t)=r0e−μt into F(t)=αKH/(rH+KH) obtains F(t)=α/[1+(r0e−μt/K)H]= α/{1+exp[Hln(r0/K)−μHt]}, which depends only on the combinations Hln(r0/K) and μH.

-l. 526. "T=2.32". What does this time correspond to in terms of cell position?

It is the position at which the number of QR.pa data points to the right equals the number of QR.p data points to the left. Quantitatively, it is 36% of the way from the V2 seam cell to the V3 seam cell (because P=Pmax−T=4.68−2.32=2.36 in units of the seam cells, where Pmax=4.68 is the position of the rightmost QR data point).

- Figure 1D. What is the GFP reporter?

The GFP report is expressed in the seam cells and the QR lineage descendants (*Pwrt-2::gfp*, transgene *heIs63*). This has been added to the figure legend.

- Figure 2-supplement 2. What statistical test has been used? I am also asking because from looking at the data it is not so clear that control and Δ intron A are really not significantly different. What is the P value?

Welch’s t-test, as in Figure 3, supplemental figure 2. The p-value is p<0.0001. This has been added to the figure legend.

- Figure 4E. The figure shows that decreasing the size ratio leads to slower mig-1 induction, but it is not clear how well it fits the data. If you split the data in Figure 4C into brackets of different ratios/rho, would it then fit the model data for the rho of that bracket? Based on Figure 4E, F, it remains open to me how well the model actually fits the data.

See new Figure 4, supplemental figure 1 for combination of experimental data and model fit.

- l.823-824. What is this a test of? I am confused because the text mentions transcripts, but the panel is about position.

We performed a Brown-Forsythe analysis on the coefficients of variation (CV) at different thresholds of mig-1 expression. See Materials and methods section “Statistical analysis of variance data” for more details.

Reviewer #2 (Recommendations for the authors):The strength of the manuscript is the overall question, the experimental data, and the simple model. But given the quality of the data, the data analysis, model fitting, and predictions could be improved or at least better presented.Specifically, I would like to see how much variation exists between different replica experiments. Right now, all the RNA-FISH data is pooled and not separated by replica experiments. One simple way to address this would be to plot the data points separated by different symbols specific to each biological replica experiment. This comment is relevant to all figures with data.

The smFISH measurements were performed in synchronized populations enriched for QR, QR.p or QR.pa, respectively. Combined, these different experiments show the full range of *mig-1* expression. The data of the individual replicate experiments of Figure 1A is shown in Figure 1, figure supplement 1.

How did the authors determine a threshold in these data sets? Applying a piecewise linear fit to the experimental data without the model enables very precise quantification of the threshold. This should be done for each replica experiment to determine the mean and standard deviation in the threshold. This should be also done for all the WT and mutant data sets to show how the expression levels and threshold are effects by different mutants.

By threshold we take the reviewer to mean the activator level K above which the *mig-1* production rate increases. A piecewise linear fit to the *mig-1* data would indeed quantify K precisely. However, a piecewise linear fit would be equivalent to taking the Hill coefficient to H→∞, which we do not want to assume. Therefore, we fit K and H simultaneously. Importantly, the value of K fit with H→∞ would be different than the value of K fit together with a finite H.

With regard to the model, the authors need to show how the model fits to the data, by overlaying the model and the data in the same plot. The authors state in the manuscript that the spatial position of the cells is proportional to time, so this should be possible. This should be done for each replica to determine the mean and standard deviation in the model parameter. The model parameters and their errors should be added to the manuscript.

We have added figures that overlay the experimental data and model fits (Figure 2, figure supplement 3 and Figure 4, figure supplement 1).

By comparing the model fits and predictions using the estimated parameter errors, authors can better determine if the differences seen in the WT and mutant experiment are significantly large to explain this difference by changing specific rates in the model.

In the majority of cases, the mutants produce changes in the data that go beyond the spreads in the data points themselves. Thus, it is unlikely that these changes could be encompassed within parameter uncertainties in the model, even if these uncertainties could be estimated from replicates. This observation justifies treating the model as a phenomenological description whose predictions manifest as one investigates changes in its parameters.

Although the authors quantify the position equals temporal variance in the mig1 expression, they do not use their model to make this prediction or even analyze the data with their model. Why is this? Can the model not compute temporal variances? Please clarify.

We now explicitly model the positive feedback and compare our findings to the experiments. Specifically, we find that when *mig-1* is subjected to feedback, both through the activator and independently from the activator, then disrupting the feedback either through loss or constitutive activation leads to increased variability in the timing of *mig-1* expression, consistent with the experiments (see the new Figure 5B, Figure 5, figure supplement 2, and Materials and methods section “Modeling the effect of feedback on timing precision”).